# Load Balancing Neurons: Controlling Firing Rates Improves Plasticity in Continual Learning

## Abstract

Neural networks in continual learning often lose plasticity: some neurons become inactive, while others fire almost constantly. This limits adaptation to shifting data and wastes capacity. Prior work mitigates this by periodically reinitializing low-utility units, but such resets can destroy previously learned features and do not proactively prevent low utility. We study a simple diagnostic measure: the firing rate of ReLU units, defined as the fraction of positive pre-activations. Low rates identify dead units, while very high rates indicate linearized, always-on units. Based on this view, we introduce a lightweight load-balancing mechanism that adjusts per-neuron thresholds to keep firing rates within a target range. Across continual ImageNet and class-incremental CIFAR-100, improvements in firing-rate distributions help explain differences in plasticity across approaches, including our load-balancing mechanism and well-known techniques, notably L2 regularization and non-affine normalization.

## 1 Introduction

Deep learning models have achieved state-of-the-art performance on a broad range of tasks when trained on fixed datasets (He et al., 2016; Liu et al., 2022). However, performance often degrades in continual learning settings, where the data distribution evolves over time (Khetarpal et al., 2022; Lee et al., 2023; Abbas et al., 2023). Simply continuing training on new data often leads to optimization instabilities and catastrophic forgetting, and the mechanisms behind these failures are not yet fully understood (Lyle et al., 2024). A common remedy is to retrain a new model from scratch, which restores accuracy but is often impractical due to computational cost.

Ideally, a continual learning system would strike a balance between *plasticity*, the ability to adapt to new data, and *stability*, the ability to preserve what it has already learned (Carpenter, 1987; Elsayed & Mahmood, 2024). Declining stability is known as *catastrophic forgetting* (McCloskey & Cohen, 1989), while declining plasticity is commonly called *plasticity loss*. Overcoming these challenges would allow models to evolve as data changes over time, mimicking the way humans and animals continuously learn without retraining from scratch.

**Contributions:**

- We introduce firing-rate analysis as a simple, interpretable diagnostic tool for neuron utilization and plasticity loss in continual learning, offering a unified view across tasks and architectures.

- We analyze and compare the firing-rate distributions of existing methods, including L2 regularization and non-affine normalization, and show how these patterns relate to plasticity loss.

- We propose a load-balancing mechanism that dynamically adjusts neuron activity by shifting activation inputs. It keeps firing rates balanced and outperforms previous baselines.

## 2 BACKGROUND

**Plasticity loss.** The phenomenon of plasticity loss has been observed in both supervised learning (Ash & Adams, 2020; Berariu et al., 2021; Dohare et al., 2024) and reinforcement learning (Lyle et al., 2022; Nikishin et al., 2022; Dohare et al., 2024). Several approaches have been proposed to improve plasticity. For instance, injecting noise during training (Ash & Adams, 2020), periodically resetting the last layers (Nikishin et al., 2022), expanding the network during training (Nikishin et al., 2023), using the Concatenated ReLU activation function Abbas et al. (2023), reinitializing low-utility neurons (Sokar et al., 2023; Dohare et al., 2024), self-normalized neuron resets based on inactivity statistics (Farias & Jozefiak, 2025), perturbing gradients of low-utility neurons Elsayed & Mahmood (2024), and regularizing weights toward their initial values Kumar et al. (2024). Some studies combine multiple techniques, for example, Lee et al. (2023) combines layer resets, Concatenated ReLUs, LayerNorm (Ba et al., 2016), and sharpness-aware minimization (Foret et al., 2021).

Many of these interventions restore plasticity by potentially destroying learned information, by resetting neurons or entire layers of the networks, rather than preventing collapse in the first place. A recent example is Self-Normalized Resets (SNR), which resets neurons that appear inactive via a self-normalized percentile test on inter-firing times (Farias & Jozefiak, 2025). In contrast, we ask how to sustain plasticity proactively through minimal architectural changes.

**Measuring neuron utility.** Many of the above methods rely on a measure of neuron utility. For example, Dohare et al. (2024) define a neuron's *contribution utility* as the magnitude of its activation multiplied by the magnitudes of all outgoing weights, tracked as a moving average during training. Sokar et al. (2023) define the *dormancy score* as the neuron's activation magnitude, normalized by the magnitudes of other neurons in the same layer and averaged over the dataset; a neuron is considered *dormant* if the dormancy score falls below a threshold. Elsayed & Mahmood (2024) define *weight utility* as the change in loss after zeroing a weight, which yields a global ranking over all weights but is more expensive to compute. SNR instead uses inactivity statistics derived from neuron firings to decide when to reset (Farias & Jozefiak, 2025).

In contrast, we introduce a firing-rate diagnostic that reveals the *distribution* of neuron utilization, across layers and over time, rather than collapsing activity into a single summary statistic.

**Continual learning benchmarks.** Many benchmarks have been proposed to evaluate continual learning, but there is no single standard, and new variants are often introduced, making comparisons difficult. In this work, we adopt the benchmarking setup of Dohare et al. (2024) to assess our method. We therefore focus on the *continual ImageNet* and *class-incremental CIFAR-100* benchmarks, which they used to evaluate continual backpropagation (CBP).

## 3 PLASTICITY THROUGH THE LENS OF FIRING RATES

Plasticity loss often reflects neuron-level collapse in ReLU networks, where some neurons stop firing and others become always active. Plasticity, in turn, depends on whether a neuron spends time on both sides of its threshold. We therefore use the *firing rate*, the fraction of pre-activations for which a neuron is active, as a diagnostic tool of neuron use. Tracking firing-rate distributions reveals training anomalies, and keeping them in a balanced range correlates with improved adaptation under distribution shift, as we will demonstrate in the experimental section.

### 3.1 PROMOTING PLASTICITY BY INCREASING PERPLEXITY

To make this intuition more precise, we now view the network through the lens of its piecewise-affine structure and its binary activation patterns. Many standard network designs, like multi-layer perceptrons and convolutional nets using ReLU or similar functions, implement a *continuous piecewise-affine (CPWA) map* that partitions the input space into regions on which the network is affine (Montúfar et al., 2014; Serra et al., 2018; Goujon et al., 2024). Each such region is determined by which hidden units are active or inactive, and there is a one-to-one correspondence between these activation patterns and the affine regions induced by the network (Zhang & Wu, 2020). This viewpoint lets us characterize the networks functional complexity in terms of how finely these regions

carve up the input space (Montúfar et al., 2014; Serra et al., 2018; Zhang & Wu, 2020); extensions to non-CPWA architectures are discussed in Appendix C.

Consider a network with parameters $\theta$ and a total of $M$ hidden units. For any input $x \in \mathcal{X}$, let $z_i(x)$ denote the pre-activation of unit $i \in \{1, \ldots, M\}$. We define the corresponding binary gate

$$S_i(x) := [z_i(x) \geq 0], \tag{1}$$

which indicates whether unit $i$ is active. The *global activation pattern* is the vector of all gates:

$$S(x) := \big(S_1(x), \ldots, S_M(x)\big) \in \{0, 1\}^M. \tag{2}$$

Each unique pattern $s \in \{0, 1\}^M$ defines a region $\mathcal{R}(s) := \{x \mid S(x) = s\}$ where the network is affine. All non-empty regions together form a partition $\mathcal{P}$ of the input space.

**Effective capacity and plasticity.** The expressive power of a CPWA network is fundamentally determined by its partition $\mathcal{P}$ (Montúfar et al., 2014; Serra et al., 2018; Takai et al., 2021). Since the network is an affine map on each region $R \in \mathcal{P}$, it cannot capture non-affine variations within a single region. Thus, to represent a nonlinear target function, the partition must separate inputs that cannot be well-approximated by the same affine function into different regions.

A naive way to quantify this capacity is to count the total number of regions $|\mathcal{P}|$, but this treats all regions equally, regardless of how often they are visited by the data. To obtain a data-dependent notion, let $\mathbf{x} \sim P_{\mathbf{x}}$ and let $\mathbf{s} := S(\mathbf{x})$ be the random activation pattern ($\mathbf{x}$, $\mathbf{s}$ are random variables; $x$, $s$ their values). We then measure the effective number of regions using the *perplexity* $2^{H(\mathbf{s})}$ of the activation patterns. Standard entropy bounds yield $2^{H(\mathbf{s})} \leq |\mathrm{supp}(\mathbf{s})| \leq |\mathcal{P}|$, where $\mathrm{supp}(\mathbf{s})$ is the support of $\mathbf{s}$, so the perplexity provides a smooth, data-dependent lower bound on the number of regions that carry non-negligible probability mass under $P_{\mathbf{x}}$. Related work has used entropy and perplexity of activation patterns to quantify how richly a network distinguishes inputs or classes (Davel et al., 2020; Hartmann et al., 2021).

Since the mapping $x \mapsto S(x)$ is deterministic, $H(\mathbf{s} \mid \mathbf{x}) = 0$; thus maximizing this lower bound is equivalent to maximizing the mutual information $I(\mathbf{x}; \mathbf{s}) = H(\mathbf{x}) - H(\mathbf{s} \mid \mathbf{x})$, thereby forcing the discrete code to retain maximal information about the input (the *InfoMax principle*; Linsker, 1988). Closely related, mutual-information objectives between activation codes and inputs have been used to explicitly increase the nonlinear expressivity of encoders (Park et al., 2021). In a continual learning setting with a time-varying distribution $P_{\mathbf{x}_t}$, maximizing $2^{H(\mathbf{s}_t)}$ continually pushes up the lower bound on $|\mathcal{P}|$. This is promoting *plasticity*, since it encourages the network to keep many different affine regions actively used, so it can keep telling apart new and changing input patterns.

### 3.2 Firing rates as an easily computable objective

The global entropy $H(\mathbf{s})$ is bounded above by the sum of the entropies of the individual units. The gap is the *total correlation (TC)* (Watanabe, 1960), which measures the redundancy (statistical dependence) among units:

$$H(\mathbf{s}) = \sum_{i=1}^{M} H(\mathbf{s}_i) - TC(\mathbf{s}). \tag{3}$$

Optimizing the joint entropy $H(\mathbf{s})$ directly is computationally challenging due to the difficulty of estimating high-dimensional dependencies (the $TC$ term). However, the first term in Equation (3), $\sum_{i=1}^{M} H(\mathbf{s}_i)$, depends only on marginal probabilities and is efficient to compute. Since each gate $\mathbf{s}_i$ is a Bernoulli variable, its entropy $H(\mathbf{s}_i)$ is maximized when its probability of being active is exactly $0.5$. Similar observations are used in binarized activation entropy regularizers that encourage individual units to be active about half the time while keeping them weakly correlated, in order to increase the joint entropy of their activation patterns (Cao et al., 2018). Therefore, maximizing the sum of marginal entropies encourages every unit to fire for half of the inputs, avoiding the "dead unit" problem and maximizing the information capacity of individual neurons.

We propose to maximize this sum as a simple, tractable stand-in for global plasticity. In principle, this objective can still leave redundancy between units (e.g., for non-zero $TC$), but we observe that making each unit active about half the time will usually also increase the joint entropy: In Section 4

we check this assumption empirically and find that the total correlation stays under control while the marginal entropies grow, so the extra unit-wise capacity corresponds to a real increase in the effective number of affine regions. A complementary view is in terms of the average Hamming distance between activation codes, which we discuss further in Appendix C.

**Firing rate.** The firing-rate perspective provides a practical way to monitor and control the marginal entropies of the gates $\mathbf{s}_i$. For each unit $i$, the gate $\mathbf{s}_i$ is a Bernoulli random variable under $\mathbf{x} \sim P_{\mathbf{x}}$, with activation probability $\Pr(\mathbf{s}_i = 1) = \Pr(z_i(\mathbf{x}) \geq 0)$. In practice, we estimate this probability from a mini-batch $x_1, \ldots, x_n$ via the *instantaneous firing rate*

$$\rho_i \coloneqq \frac{1}{n} \sum_{j=1}^{n} [z_i(x_j) \geq 0], \tag{4}$$

which is the empirical mean of the gate $\mathbf{s}_i$ over the batch. To smooth batch noise and remain responsive to a changing data distribution, we track an *exponentially averaged firing rate* with decay $\tau \in [0, 1]$, denoted $\bar{\rho}_i$, to capture the longer-term trend:

$$\bar{\rho}_i \leftarrow \tau \bar{\rho}_i + (1 - \tau)\rho_i. \tag{5}$$

For ReLU units, the pre-activation $z_i$ induces two regimes: an *off* regime $z_i < 0$ (zero output and zero slope) and an *on* regime $z_i \geq 0$ (positive output and unit slope). The firing rate $\bar{\rho}_i$ thus directly reflects how often unit $i$ is in each regime. Near $\bar{\rho}_i = 0$ the neuron is *dead* or *dormant* and contributes neither activation nor gradient. Near $\bar{\rho}_i = 1$ it behaves as an almost linear pass-through with gradients still flowing, but adding no curvature. In both extremes, the entropy of the gate $\mathbf{s}_i$ is close to zero, and the unit contributes little to the global entropy $H(\mathbf{s})$, reducing effective capacity and plasticity. This view motivates using firing rates both as a diagnostic for plasticity and as a control signal for the load-balancing mechanism introduced later.

### 3.3 Firing-rate dynamics in continual benchmarks

In this subsection, we study firing-rate behavior on the continual ImageNet and class-incremental CIFAR-100 benchmarks. These settings build on widely used datasets, are tailored to CNN and ResNet architectures that provide a stronger test of plasticity than simpler MLP models, and are straightforward to interpret. Together, they offer a useful testbed for examining how neuron firing rates evolve under distribution shift.

**Continual ImageNet: firing-rates and accuracy.** In the continual ImageNet benchmark (Dohare et al., 2024), the model is trained on a sequence of binary classification tasks. For each task it learns to discriminate two randomly sampled classes from ImageNet (Deng et al., 2009) before moving on to a new pair. Because all tasks have comparable difficulty, accuracy degradation across tasks indicates reduced plasticity (Dohare et al., 2024). Following Dohare et al. (2024), images are downsampled to $32 \times 32$ resolution, which preserves ImageNet's class diversity while making long sequential runs computationally feasible for thousands of tasks. This setup is substantially more demanding than MNIST-scale benchmarks. We use a compact CNN with ReLU activations (see Appendix D for training details).

Firing-rate distributions and test accuracies of several baselines are shown in Figure 1. Hyperparameter details are provided in Appendix D. Accuracy curves report the mean over multiple seeds, with shaded bands denoting $\pm 1$ standard error of the mean. Because tasks are formed by randomly sampled binary class pairs, this benchmark exhibits higher run-to-run variability. The error bands make this variability explicit without altering the overall ranking among methods, and results are best interpreted in terms of the trajectory of accuracy curves, which captures plasticity dynamics. Absolute accuracies for all baselines and tasks are listed in Appendix E.

Throughout this paper, firing-rate distributions are visualized as two-dimensional heatmaps: the horizontal axis denotes training progress (number of tasks), the vertical axis spans firing rates from 0 to 100%, and each column shows a histogram over all neurons in the network. Brighter regions indicate many neurons with that firing rate, whereas darker regions indicate fewer. These plots reveal how activity patterns evolve over time.

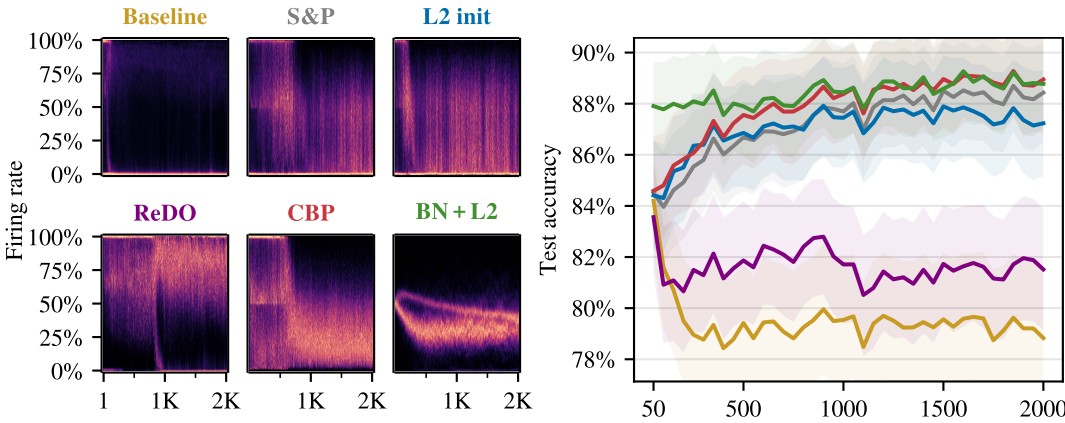

Figure 1: **Continual ImageNet:** Evolution of firing rate distributions (left) and test accuracy with error bands (right), averaged over 10 seeds. Key findings are summarized in the main text.

Key findings:

1. **Baseline.** Accuracy degrades rapidly over tasks. Firing rates collapse toward $\approx 0\%$ for most neurons, aligning with the drop in accuracy.

2. **S&P.** Shrink-and-perturb combines L2 regularization with small parameter noise (Ash & Adams, 2020). It increases the fraction of active neurons (up to $\approx 75\%$) and substantially improves accuracy, though many neurons remain near $\approx 0\%$.

3. **L2 init.** L2 init regularizes weights toward their initialization (Kumar et al., 2024). Like S&P, it yields higher activity (up to $\approx 80\%$) and higher accuracy, while a subset still collapses toward $\approx 0\%$.

4. **ReDO.** Recycling dormant neurons resets neurons that fall below a dormancy threshold (Sokar et al., 2023). It largely eliminates the $\approx 0\%$ regime, but leads to widespread saturation early ($\approx 100\%$) and sustained high firing ($\approx 75-95\%$), consistent with linearized ReLUs and reduced accuracy.

5. **CBP.** Continual backpropagation resets "underutilized" neurons using an activation-magnitude criterion (Dohare et al., 2024). It largely prevents collapse to $\approx 0\%$, but early saturation near $\approx 100\%$ again suggests linearized ReLUs. After $\approx 500$ tasks, the distribution recenters and broadens, coinciding with improved accuracy.

6. **BatchNorm + L2.** Combining BatchNorm (Ioffe & Szegedy, 2015) with L2 regularization already prevents extreme firing rates. Both accuracy and plasticity stay high.

**Class-incremental CIFAR-100: firing-rates and accuracy.** In the class-incremental CIFAR-100 benchmark, the model begins by learning to classify a small set of randomly sampled classes fomr CIFAR-100 (Krizhevsky et al., 2009). Additional classes are gradually introduced, and the model must classify all classes encountered so far until the full set of 100 is included. Because the classification problem grows harder as more classes accumulate, performance can decline even without plasticity loss (Dohare et al., 2024). To disentangle this effect, we compare continual training with a model trained from scratch on the same subset. If the continually trained model performs worse than a model trained from scratch, this indicates plasticity loss, whereas superior performance signals knowledge transfer (Dohare et al., 2024). We use a ResNet-18 with ReLU activations (see Appendix D for training details).

Firing-rate distributions and test accuracies of several baselines are shown in Figure 2. Results are averaged over multiple seeds, with shaded bands denoting $\pm 1$ standard error of the mean.

Key findings:

1. **Baseline.** Neurons gradually become inactive, though less severely than continual ImageNet. Many units reach $\approx 0\%$ firing, limiting the capacity for new tasks.

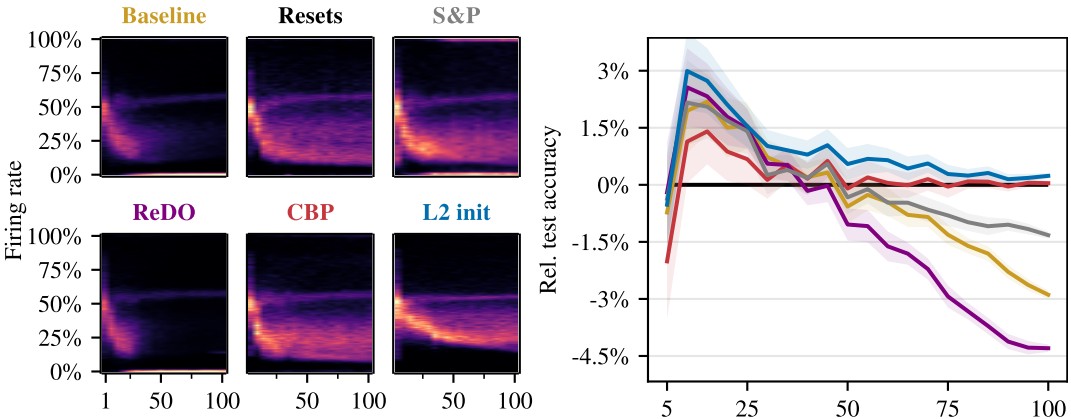

Figure 2: **Class-incremental CIFAR-100:** Firing-rate distributions (left) and test accuracy with error bands (right), averaged over 15 seeds. Key findings are summarized in the main text.

2. **Resets.** Reinitializing the full network after each task prevents collapse and preserves adaptability. As tasks become harder, the firing-rate distribution shifts accordingly.

3. **S&P.** Firing-rate distributions move toward the *Resets* regime, but many neurons still cluster near $\approx 0\%$ (dead) and $\approx 100$ (effectively linearized). This imbalance corresponds to lower accuracy than *Resets*.

4. **ReDO.** Like the baseline, most neurons become inactive; accuracy is even lower than the baseline.

5. **CBP.** Firing-rate distributions closely match *Resets*, and test accuracy is comparable. Most units lie around 10-30%, with some up to $\approx 60\%$, indicating sustained plasticity.

6. **L2 init.** Firing rates concentrate even more tightly around $\approx 50\%$ than with CBP. It achieves the highest accuracy among these baselines.

**Summary of findings.** In summary, firing-rate analysis provides a straightforward lens on plasticity in continual learning. Across both benchmarks, conventional training rapidly leads to inactive neurons, while CBP and L2 init maintain balanced firing-rate distributions that closely resemble those of fully reset networks.

## 4 LOAD BALANCING: BOUNDING PER-NEURON FIRING RATES

In this section, we investigate whether explicit control of firing rates can achieve performance beyond full network resets between tasks. We introduce a lightweight mechanism, *load balancing*, that shifts each neuron's pre-activation to keep its long-term firing rate within target bounds. Concretely, before a ReLU we replace the pre-activation $x$ by a shifted value

$$\tilde{x} = x + \text{sg}(\beta), \tag{6}$$

where $\beta \in \mathbb{R}$ is a per-neuron offset initialized to zero and $\text{sg}(\cdot)$ denotes a stop-gradient (i.e., $\beta$ is treated as constant for backpropagation). Rather than updating $\beta$ by gradient descent, we adjust it after each batch using a simple controller based on the exponentially averaged firing rate $\bar{\rho}$:

$$\beta \leftarrow \begin{cases} \beta + \alpha & \text{if } \bar{\rho} < \rho_{\min} \\ \beta - \alpha & \text{if } \bar{\rho} > \rho_{\max} \\ \beta & \text{otherwise,} \end{cases} \tag{7}$$

where $\alpha > 0$ is a small step size (we use $\alpha = 0.001$) and $[\rho_{\min}, \rho_{\max}] \subset [0, 1]$ specifies the target range. Because the update mechanism uses feedback from measured firing rates, we compute those rates on the shifted pre-activations $\tilde{x}$, which is the signal the ReLU actually receives. During inference, $\beta$ is fixed. Intuitively, this shifts the effective activation threshold horizontally so that neurons spending too little (or too much) time in the active regime are nudged toward the target range.

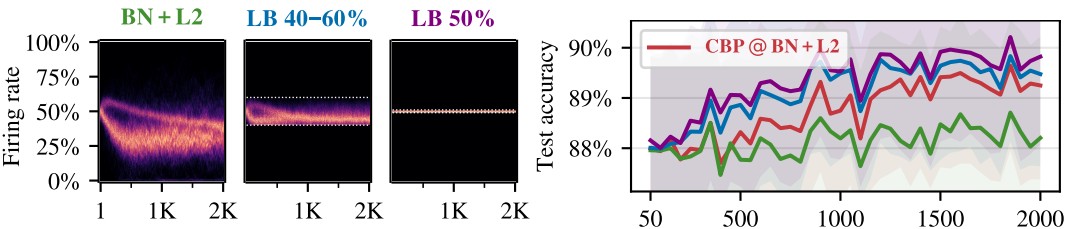

Figure 3: **Continual ImageNet:** Firing-rate distributions (left) and test accuracy (right) under BN + L2 with and without load balancing. Constraining neurons to intermediate regimes (40-60% or 50%) improves accuracy over both the unconstrained baseline and CBP.

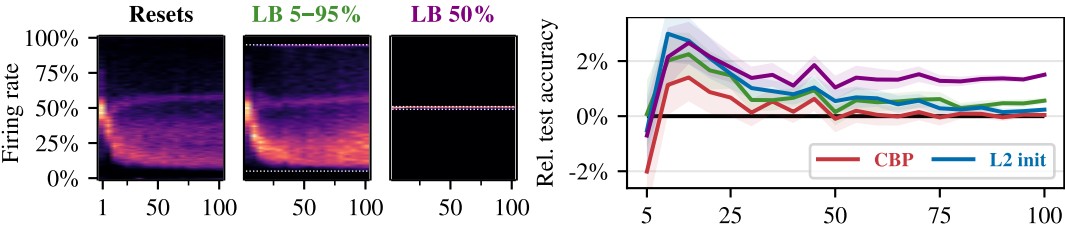

Figure 4: **Class-incremental CIFAR-100:** Firing-rate distributions (left) and test accuracy (right) for different load-balancing ranges. Loose bounds (5-95%) reproduce reset-like behavior with improved accuracy, while control near 50% yields further gains, surpassing resets and prior methods.

**Practical considerations.** In practice, the shift $\beta$ can compete with the learned bias of the preceding layer. We mitigate this by placing a BatchNorm immediately before the shift and using L2 regularization. This configuration is already present in the class-incremental CIFAR-100 setup; for continual ImageNet we adopt the BN + L2 configuration. We implement load balancing as a small layer inserted before each activation, which measures firing rates and adapts the shifts. PyTorch code for this layer is provided in Appendix H.

**Load balancing in continual ImageNet.** We use BN + L2 as the base configuration, since load balancing is more stable in the presence of normalization. For comparability, we re-tune all baselines on this same setup; results are shown in Figure 8, and hyperparameter details and full tables are provided in Appendices D and E. We denote methods instantiated on this base as "@ BN + L2".

On continual ImageNet, we evaluate load balancing under different firing rate bounds (Figure 3). Constraining firing rates to an intermediate band (40–60%) yields higher accuracy than both unconstrained BN + L2 and CBP @ BN + L2, and fixing all neurons to 50% performs even better.

**Load balancing in class-incremental CIFAR-100.** In the class-incremental CIFAR-100 benchmark, we evaluate two load-balancing configurations and compare their performance relative to the Resets baseline, where the network is reinitialized after each task. The results are presented in Figure 4. Using loose bounds of 5-95% reproduces firing-rate patterns similar to the reset-based baseline. This configuration already achieves higher accuracy than both resets and CBP. Pushing the bounds more aggressively to 50% yields even better performance.

**Measuring entropy.** As discussed in Section 3.2, the joint entropy of the binary gates decomposes as $H(\mathbf{s}) = \sum_{i=1}^{M} H(\mathbf{s}_i) - TC(\mathbf{s})$. To monitor how the load-balancing mechanism affects these terms, we evaluate them on the test set after each task, using deterministic evaluation (e.g., BatchNorm in inference mode). For each unit $i$, we estimate the marginal activation probability $\Pr(\mathbf{s}_i = 1)$ from the empirical firing rate on the test inputs, and compute the corresponding Bernoulli entropy $H(\mathbf{s}_i)$. We report the sum $\sum_{i=1}^{M} H(\mathbf{s}_i)$ as a measure of the total marginal capacity.

Directly estimating the total correlation $TC(\mathbf{s})$ is intractable for high-dimensional $\mathbf{s}$, as it would require the full joint distribution over $\{0, 1\}^M$. We therefore adopt a Chow–Liu tree approximation

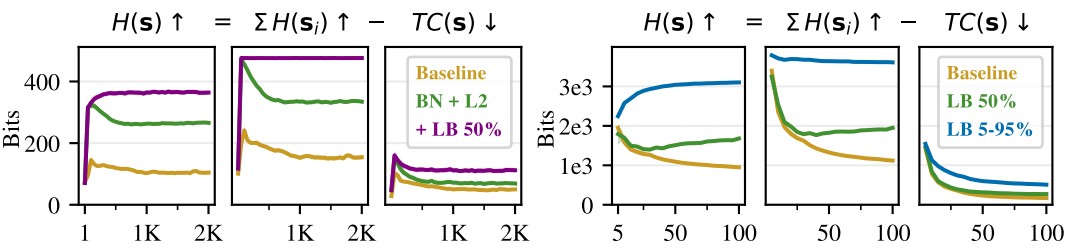

Figure 5: **Entropy:** Continual ImageNet (left) and class-incremental CIFAR-100 (right), with $H(\mathbf{s})$, $\sum_i H(\mathbf{s}_i)$, and $TC(\mathbf{s})$ computed on the test set after each task and averaged over 3 random seeds.

(Chow & Liu, 1968). Concretely, we first estimate the pairwise mutual information $I(\mathbf{s}_i; \mathbf{s}_j)$ for all pairs $(i, j)$ from empirical joint activation counts. We then construct a maximum spanning tree (MST) over the $M$ units, using these mutual informations as edge weights. For a distribution that factorizes according to this tree-structured graphical model, the total correlation equals the sum of mutual informations along the tree edges,

$$TC(\mathbf{s}) = \sum_{(i,j) \in \text{MST}} I(\mathbf{s}_i; \mathbf{s}_j), \tag{8}$$

and we use this quantity as a tractable approximation that captures the dominant pairwise dependencies while ignoring higher-order interactions.

The resulting trajectories of $\sum_{i=1}^{M} H(\mathbf{s}_i)$ and the Chow-Liu estimate of $TC(\mathbf{s})$ are shown in Figure 5. With load balancing, the sum of marginal entropies increases substantially across tasks, whereas the total-correlation term grows only mildly. Consequently, the estimated joint entropy $H(\mathbf{s})$ increases, indicating that the additional unit-wise capacity translates into a genuine increase in the effective number of active affine regions rather than into redundant, highly correlated gates.

**Summary of findings.** This section shows that explicitly controlling firing rates with load balancing improves continual-learning performance. Constraining rates to intermediate ranges, e.g., 40–60% or a fixed 50%, yields higher accuracy than both the reset-based baseline and CBP on the two benchmarks. These settings maintain diverse activation patterns and align with the earlier argument that a 50% firing rate maximizes activation-state entropy and supports nonlinearity. Consistent with this, our Chow-Liu estimates of total correlation show that load balancing substantially increases the sum of marginal entropies while only mildly increasing redundancy, so the joint entropy $H(\mathbf{s})$ and the effective number of active regions both grow across tasks. Taken together, the results highlight balanced neuron utilization as a useful target and indicate that firing-rate control is a practical mechanism for improving plasticity and knowledge transfer.

## 5 FURTHER EXPERIMENTS

### 5.1 BEYOND RELU

The main focus of this paper is on ReLU activations. Nonetheless, it is also natural to ask how controlling the firing rate affects other activation functions. We applied load balancing to LeakyReLU (Maas et al., 2013) and GELU (Hendrycks & Gimpel, 2016). The results are shown in Figures 6 and 9. In each figure, the left panels show the firing-rate distributions without load balancing, while the right panels compare performance without load balancing (solid lines) and with load balancing at 50% (dashed lines).

For the class-incremental CIFAR-100 in Figure 6 we see that the firing rates for all three activation functions show a loss of plasticity. Accordingly, all three profit from load balancing at 50% as can be seen in the right panel. Curiously, the ReLU activation function profits most (see right edge of the accuracy panel): without load balancing it has the worst performance, with 50% load balancing it has the best (together with load balanced LeakyReLU).

For continual ImageNet, we see from the firing rates distribution that loss of plasticity is not a big issue for any of the three activation functions (left panels of Figure 9). As discussed earlier, this

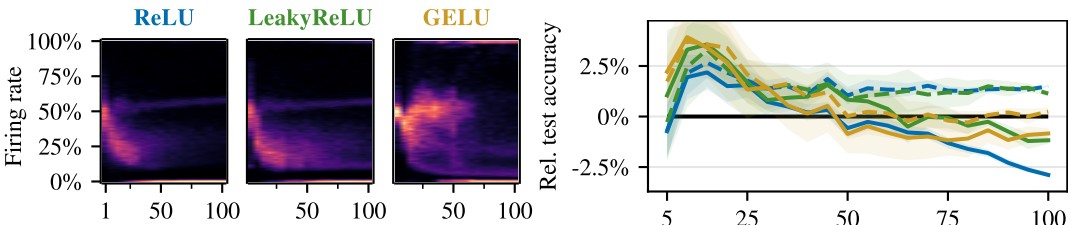

Figure 6: **Class-incremental CIFAR-100:** Firing-rate distributions (left) and test accuracy (right) for ReLU, LeakyReLU, and GELU, all based on the BN + L2 setup. In the right plot the dashed lines show the results for load balancing 50%.

is due to the use of BatchNorm and L2 regularization for this setup. It appears that all activation functions profit from load balancing, but the large variances (the background is completely covered greenish), makes the continual ImageNet experiment somewhat inconclusive.

## 5.2 REINFORCEMENT LEARNING

Beyond supervised continual learning, we also tested our load-balancing mechanism in a reinforcement-learning setting (see Appendix B.1 for details). We augmented Rainbow DQN (Hessel et al., 2018) on the Atari-5 benchmark (Aitchison et al., 2023) and *DemonAttack* (Sokar et al., 2023) with load-balancing layers on top of a LayerNorm baseline, and evaluated performance using human-normalized scores. Across the 6 games, the LayerNorm + load-balancing agent outperforms the LayerNorm baseline in 4 games and matches it in the remaining 2, while the original Rainbow baseline is worse than the load-balancing variant in 5 of 6 games. Firing-rate distributions in the appendix mirror these trends, showing that load balancing prevents severe activation collapse and keeps neurons operating near the desired 50% firing rate.

## 5.3 CASE STUDY: DESIGNING NETWORKS FOR BALANCED FIRING RATES

The preceding sections showed that plasticity correlates with non-collapsed firing-rate distributions, with the best results near 50%. While load balancing enforced this explicitly, we now ask whether architecture alone can induce the same behavior. Guided by firing-rate diagnostics, tracking layer-wise distributions, we identify minimal, effective modifications. The goal is not exhaustive architecture design, but a targeted case study of how the firing-rate lens translates into practical design choices.

**Non-affine normalization.** A direct way to keep ReLU units near the 50% firing regime is to keep their pre-activations centered at the threshold (zero). Normalization layers compute a standardized signal and then apply a learned affine transform with parameters $\gamma$ and $\alpha$:

$$\tilde{x} = \frac{x - \mathbb{E}[x]}{\sqrt{\mathrm{Var}(x) + \epsilon}} \cdot \gamma + \alpha. \qquad (9)$$

BatchNorm (Ioffe & Szegedy, 2015) estimates the mean and variance over the batch; LayerNorm (Ba et al., 2016) does so over features. Crucially, the learned shift $\alpha$ can move the normalized signal away from zero, so these layers generally do not keep the activation centered at the ReLU threshold.

Therefore, we consider non-affine variants that drop the learned scale $\gamma$ and shift $\alpha$. As shown in Figure 10, non-affine BatchNorm + L2 yields firing rates tightly clustered around $\approx 50\%$ and matches the performance of CBP. In contrast, non-affine LayerNorm + L2 produces firing rates broadly distributed across 0-100% and achieves the highest accuracy on this benchmark, surpassing even our load balancing results. This indicates that while centering promotes balanced activity, performance gains can also arise from wider, layer-wise variability in firing rates.

Adding load balancing that targets a strict 50% firing rate on top of non-affine LayerNorm reduces accuracy relative to non-affine LayerNorm alone (see Figure 10). This suggests that enforcing a single target rate across all units is not universally optimal; instead, balanced utilization distributed across the full range of firing rates, varying by layer and unit, appears beneficial. In Figure 7, non-affine BatchNorm also improves accuracy on class-incremental CIFAR-100 and already exceeds

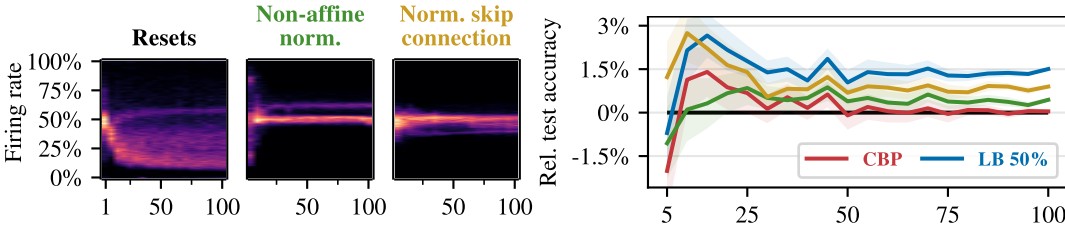

Figure 7: **Class-incremental CIFAR-100:** Firing-rate distributions (left) and test accuracy (right) under non-affine normalization and normalized skip connections. Both modifications steer firing rates toward balanced regimes and improve accuracy beyond resets and CBP.

the reset-based baseline and CBP. The corresponding firing-rate distributions are concentrated in an intermediate range rather than collapsing to 0% or saturating near 100%.

**Normalized skip connections.** Building on the normalization view, Figure 11 plots the firing rates of the first and second ReLU in each ResNet block for class-incremental CIFAR-100. Despite using non-affine BatchNorm, the distribution of the second ReLU is skewed (second panel). The cause is architectural: following the classical post-activation block of He et al. (2016) as used by Dohare et al. (2024) (`Conv-BatchNorm-ReLU-Conv-BatchNorm`, skip connection, second ReLU), the pre-activation to the second ReLU is *not* normalized. In the "Norm. skip connection" variant, we apply a non-affine BatchNorm to the skip path. The right panels of Figure 11 show that this yields a more symmetric distribution around $50\%$. Figure 7 confirms the same trend network-wide together with a corresponding gain in accuracy. While it is known that the original post-activation block has drawbacks, our results provide a complementary explanation through the lens of firing-rate balance.

**Summary of findings.** Non-affine normalization layers and normalized skip connections shift firing-rate distributions toward balanced regimes, often near 50%, and consistently improve accuracy in continual benchmarks. While their benefits have been discussed from other perspectives, viewing them through firing-rate dynamics highlights why such modifications improve performance.

## 6 LIMITATIONS

**Load balancing beyond the ReLU family.** For the ReLU activations (and to some degree also for LeakyReLU and GELU, see Section 5.1) the load is easily definable via the sign of the pre-activation. However, for saturating symmetric activations such as tanh or sigmoid it is less clear how the load should be measured. Options include controlling the amount of saturation (large absolute pre-activations) and the degree of linearity (pre-activations around zero). We leave it to future work to link the entropy-based view of the binary abstraction we pursued to these cases.

**Beyond normalization layers.** Our load-balancing layer is designed to operate in conjunction with normalization layers, which stabilize pre-activations so that firing-rate shifts are meaningful. Methods like CBP are more agnostic to the underlying architecture. Nonetheless, load balancing remains simple to implement, has minimal overhead, and integrates naturally into widely used architectures such as ResNets.

## 7 CONCLUSION

This work identifies neuron firing rates as a simple, interpretable signal for diagnosing and improving plasticity in continual learning. We show that continual training often drives rates to extremes, producing dead or effectively linear units. To counter this, we introduce a lightweight load-balancing mechanism that maintains firing rates within a target range and improves performance on two established benchmarks, surpassing reset-based baselines in some cases. We further demonstrate that architectural choices, such as non-affine normalization and normalized skip connections, implicitly encourage similar balance. Taken together, these results position activation statistics as a practical design tool for sustaining plasticity and building more robust continual learners.

## USE OF LARGE LANGUAGE MODELS (LLMs)

Large language models (LLMs) were used in the preparation of this paper as a general-purpose writing assistant. They helped with phrasing, improving clarity and flow of the manuscript, and refining figure captions. In addition, LLMs were occasionally consulted for expository assistance on background concepts, such as entropy and mutual information, but all theoretical contributions, research ideas, experimental design, and analysis originated from the authors. No parts of the paper were generated solely by an LLM without subsequent author revision and verification.

## REPRODUCIBILITY STATEMENT

Our experiments build on the official implementation of Dohare et al. (2024)[1], using the same hyperparameters as reported in their paper. We make two deviations. First, we add our load-balancing layer before each ReLU activation (implementation details are in Appendix H). Second, when using L2 regularization we apply it to weights only, excluding biases and normalization-layer parameters (see Appendix D for more details). All new hyperparameters are specified in the main text.

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

# A  ADDITIONAL FIGURES

This section collects supplementary visualizations referenced in the main text.

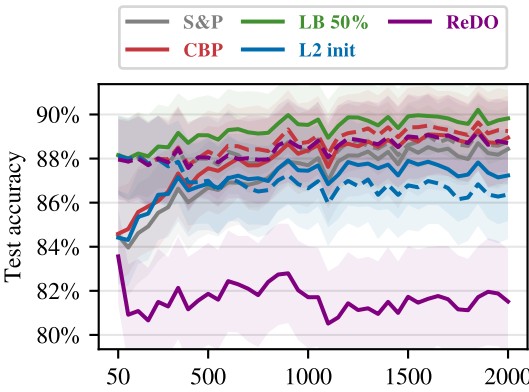

Figure 8: **Continual ImageNet:** Test accuracy of additional baselines. Dashed lines indicate the variants based on the "BN + L2" configuration.

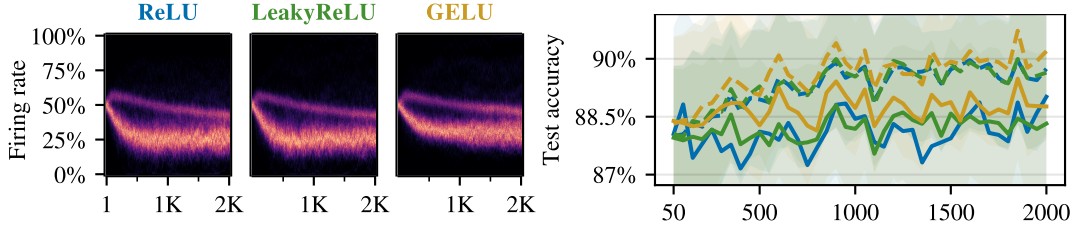

Figure 9: **Continual ImageNet:** Firing-rate distributions (left) and test accuracy (right) for ReLU, LeakyReLU, and GELU, all based on the BN + L2 setup. In the right plot the dashed lines show the results for load balancing at 50%.

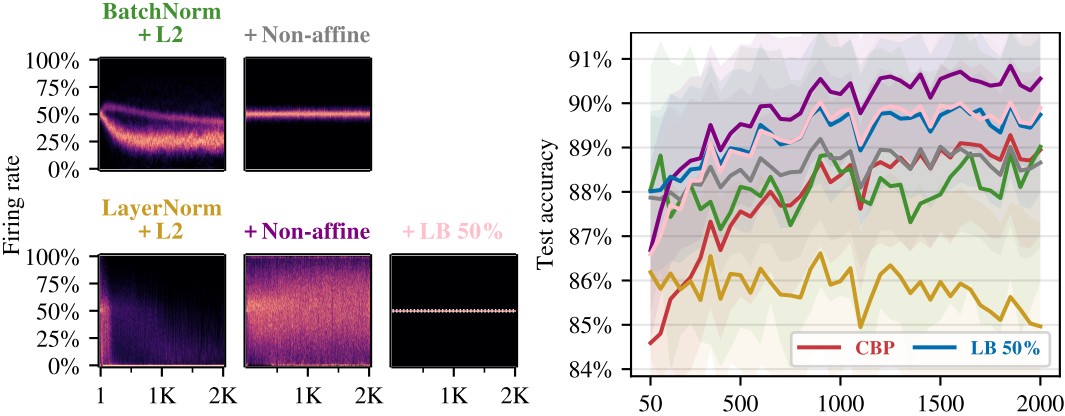

Figure 10: **Continual ImageNet:** Effects of non-affine normalization on firing-rate distributions (left) and test accuracy (right). Non-affine BatchNorm + L2 clusters firing rate near 50%, while non-affine LayerNorm + L2 broadens the distribution across layers. Both configurations improve accuracy, with non-affine LayerNorm achieving the highest performance of all methods. Enforcing 50% via load balancing on top of non-affine LayerNorm tightens rates but lowers accuracy.

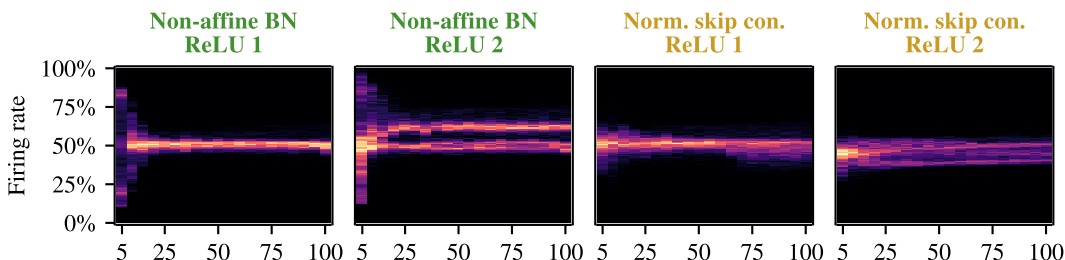

Figure 11: **Class-incremental CIFAR-100:** Firing-rate distributions of the first and second ReLU within all ResNet blocks. **Left:** Non-affine BatchNorm baseline; the second ReLU distribution is skewed. **Right:** Non-affine BatchNorm on the skip path before addition re-centers the second ReLU pre-activation, yielding a more symmetric distribution around 50%.

# B    MORE EXPERIMENTS

## B.1    REINFORCEMENT LEARNING

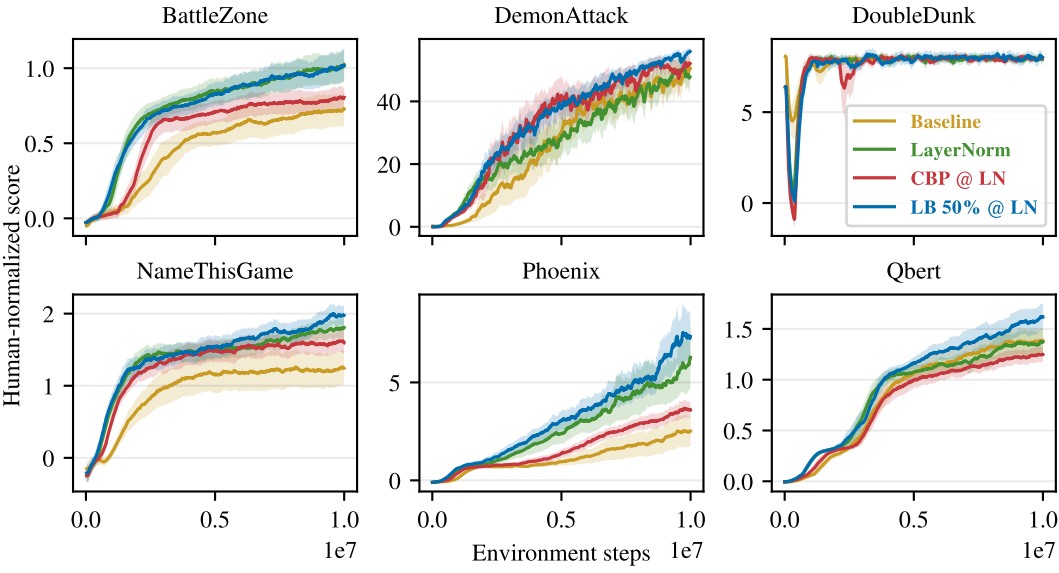

Figure 12: **Rainbow DQN:** Human-normalized scores (0 = random, 1 = human) on 6 Atari games for four variants: the original baseline, a baseline with LayerNorm, LayerNorm + load balancing targeting 50% firing rate, and LayerNorm + CBP. Curves show the mean over 5 random seeds and are smoothed with a time-weighted exponential moving average.

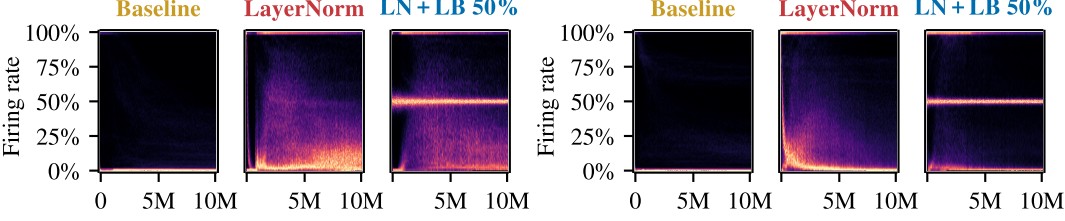

Figure 13: **Rainbow DQN:** Firing-rate distributions of BattleZone (left) and DemonAttack (right).

We conducted reinforcement learning experiments with Rainbow DQN (Hessel et al., 2018) on the Atari-5 benchmark (Aitchison et al., 2023) and on *DemonAttack*, an environment previously used to demonstrate plasticity loss (Sokar et al., 2023). We consider four agents: (i) the original Rainbow DQN baseline, (ii) Rainbow DQN with LayerNorm in the hidden layers, (iii) the same architecture further augmented with our load-balancing layers targeting a 50% firing rate (LN + LB 50%), and (iv) LayerNorm with CBP. We found that disabling load balancing for the value head consistently improved performance, so all reported results use this configuration. Other than these architectural changes, we keep the Rainbow hyperparameters fixed and use the Adam optimizer. Performance is reported in terms of *human-normalized scores* (HNS), computed as

$$\text{HNS} = \frac{\text{score}_{\text{agent}} - \text{score}_{\text{random}}}{\text{score}_{\text{human}} - \text{score}_{\text{random}}}, \tag{10}$$

so that 0 corresponds to a random agent and 1 to a human expert. Across the 6 games, the load-balancing agent achieves higher human-normalized scores than the LayerNorm baseline in 4 games and matches it in the remaining 2, as shown in Figure 12; the original Rainbow baseline is competitive in some games (on par with both agents in *DoubleDunk*, on par with LayerNorm in *DemonAttack*, and better than LayerNorm in *Qbert*), but worse than load balancing in 5 of 6 games. Load balancing outperforms CBP in all 6 games.

The firing-rate distributions in Figure 13 mirror these performance trends. In the original Rainbow baseline, almost all neurons fire at very low rates, indicating severe collapse. Adding LayerNorm partially alleviates this, but most firing rates still remain far below 50%. With load balancing, by contrast, most neurons in the hidden layers operate close to the 50% target (the value head is excluded by design). The distributions also clarify why load balancing helps in *DemonAttack* but not in *BattleZone*: in *BattleZone*, the LayerNorm baseline already causes more neurons to fire more frequently, leaving less room for improvement, whereas in *DemonAttack* the LayerNorm baseline remains highly collapsed and load balancing produces a much larger shift in firing rates.

## C  EXTENDED THEORETICAL DISCUSSION

**Beyond continuous piecewise-affine networks.**   When a network is built from affine layers together with piecewise-linear operations (e.g., ReLU and max-pooling), the overall input-output map is continuous piecewise affine (CPWA), so activation patterns index affine regimes of the computation (Montúfar et al., 2014; Serra et al., 2018; Goujon et al., 2024). This includes common CNNs and ResNets with BatchNorm at inference time: with frozen running statistics, BatchNorm reduces to an affine transformation and therefore preserves piecewise linearity structure (Bunel et al., 2018; Balestriero & Baraniuk, 2022). In architectures employing smooth normalizations such as Layer-Norm, the map is no longer piecewise affine, since LayerNorm normalizes by a per-example standard deviation (involving a square root and division), introducing a smooth nonlinearity even if the surrounding layers are linear (Ba et al., 2016; Ni et al., 2024). In that case, $S$ should be interpreted as indexing *operating regimes* or effective subnetworks (which ReLU branches are taken, and hence which local computation is realized) rather than exact affine regions. Nonetheless, the coding viewpoint remains useful: entropy or mutual-information objectives over activation codes are motivated as a way to encourage diverse, noise-robust internal regimes independent of strict CPWA assumptions (Park et al., 2021).

**Average Hamming distance.**   Here we make explicit how our marginal-entropy objective relates to the average Hamming distance between binary activation codes. Let $p_i := \Pr(\mathbf{s}_i = 1)$, and consider two independent inputs $x, x' \sim P_{\mathbf{x}}$ with codes $S(x)$ and $S(x')$. Their expected Hamming distance decomposes as

$$\bar{d}_H := \mathbb{E}_{x,x'}\big[d_H\big(S(x), S(x')\big)\big] = \sum_{i=1}^{M} \Pr\big(\mathbf{s}_i(x) \neq \mathbf{s}_i(x')\big) = \sum_{i=1}^{M} 2\, p_i(1 - p_i). \tag{11}$$

Both the Bernoulli entropy $H(\mathbf{s}_i)$ and the variance term $p_i(1 - p_i)$ are maximized at $p_i = 1/2$, and are strictly decreasing as $p_i$ moves towards 0 or 1. Thus, maximizing $\sum_i H(\mathbf{s}_i)$ not only avoids dead or saturated units, but also increases the average Hamming distance $\bar{d}_H$ between activation patterns, making codes for different inputs more widely separated. This perspective is closely related to the

noisy-channel analysis of Park et al. (2021), who show that in their Neural Activation Coding objective the mutual information between inputs and noisy binary codes lower-bounds the average Hamming distance between codewords, so maximizing mutual information yields maximally separated, noise-robust activation codes.

## D    TRAINING DETAILS

**Continual ImageNet.**   The model is a compact CNN consisting of three convolutional layers followed by two fully connected layers with ReLU activations. L2 regularization is applied to weights only, excluding biases and parameters of normalization layers. This differs from the original CBP codebase but follows common practice and was necessary for strong L2 performance. All reported results are averaged over 10 random seeds. All experiments were run on a single NVIDIA A100 GPU and a full run takes approximately 2.5 hours per seed.

All hyperparameters and sweep ranges are reported in Table 1; set-valued entries indicate the values considered in our sweeps. We tune two *reference configurations* that serve as starting points for subsequent experiments: **Baseline** (no BatchNorm, no L2) and **BN+L2** (both enabled). For normalization- and regularization-based additions (*BN*, *L2*, *S&P*, *L2 init*, and their combinations), we jointly sweep the learning rate and the relevant regularization hyperparameters (L2 strength where applicable, and noise scale for L2 init), selecting the setting that maximizes accuracy averaged over the final 100 tasks. The best-performing hyperparameters are:

- **Baseline:** learning rate 0.001
- **BN only:** learning rate 0.01
- **L2 only:** learning rate 0.03, L2 strength $3e{-}06$
- **S&P:** learning rate 0.01, L2 strength $1e{-}05$, noise scale $1e{-}05$
- **S&P + BN:** learning rate 0.03, L2 strength $1e{-}06$, noise scale $1e{-}06$
- **L2 init:** learning rate 0.01, L2 init strength $1e{-}05$
- **L2 init + BN:** learning rate 0.03, L2 init strength $1e{-}05$
- **BN + L2:** learning rate 0.03, L2 strength $1e{-}06$

For reset-based baselines that add their own hyperparameters (*ReDO*, *CBP*, *Load balancing*), we instantiate them on top of a tuned reference configuration and keep the reference hyperparameters fixed (learning rate and, if present, L2 strength). We then sweep only the method-specific hyperparameters and again select by accuracy over the last 100 tasks.

The reset-based baselines (*ReDO*, *CBP*), and *Load balancing* introduce method-specific hyperparameters. To separate these from the choice of optimizer-level hyperparameters (learning rate, L2 strength), we select two *reference configurations* that serve as fixed backbones for these methods: **Baseline** (no BatchNorm, no L2) and **BN + L2** (both enabled). We keep the reference hyperparameters fixed, sweeping only method-specific hyperparameters and again selecting by accuracy over the final 100 tasks. The best-performing hyperparameters are:

- **ReDO @ Baseline:** dormancy threshold 0.1, recycling period 10
- **ReDO @ BN + L2:** dormancy threshold 0, recycling period 10
- **CBP @ Baseline:** replacement rate 0.0003
- **CBP @ BN + L2:** replacement rate 0.0003
- **Load balancing @ BN + L2:** target range 50%

For the LeakyReLU and GELU ablations, we additionally sweep the learning rate and find the same optimal learning rate as with ReLU.

**Class-incremental CIFAR-100.**   Each task begins with five randomly sampled classes from CIFAR-100. Every 200 epochs, five additional classes are introduced, and the model is trained to classify all classes encountered so far. The architecture is a ResNet-18 with ReLU activations,

adapted to $32 \times 32$ CIFAR-100 images. Each new task starts from the best checkpoint of the previous task, selected using validation accuracy, effectively applying early stopping (Dohare et al., 2024). All reported results are averaged over 15 random seeds. All experiments were run on a single NVIDIA A100 GPU and a full run takes approximately 6 hours per seed.

All hyperparameters and sweep ranges are reported in Table 2. Across all methods, we use the same base learning rate and learning-rate schedule as Dohare et al. (2024). For CBP and S&P, we use the hyperparameters reported by Dohare et al. (2024). We found that L2 init performs better when combined with usual L2 regularization (toward zero). The remaining best-performing hyperparameters are:

- **L2 init:** L2 init strength 0.0001
- **ReDO:** dormancy threshold 0.1, recycling period 100
- **Load balancing:** target range 50%

**Rainbow DQN.** We use the CleanRL codebase (Huang et al., 2022) for our Rainbow DQN implementation. We did not change any of the hyperparameters. All results are averaged over 5 random seeds. All experiments were run on a single NVIDIA A100 GPU and a full run takes approximately 15 hours per seed.

Table 1: Hyperparameters for continual ImageNet.

| Hyperparameter | Value |
|---|---|
| Number of tasks | 2000 |
| Epochs per task | 250 |
| Early stopping | No |
| Batch size | 100 |
| Optimizer | SGD |
| Learning rate | $\{0.001, 0.003, 0.01, 0.03, 0.1\}$ |
| L2 strength | $\{1e{-}06, 3e{-}06, 1e{-}05, 3e{-}05\}$ or 0 |
| Learning rate schedule | None |
| Momentum | 0.9 |
| *L2 init* strength | $\{0.01, 0.001, 0.0001, 0.00001\}$ |
| *S&P* noise scale | $\{1e{-}07, 1e{-}06, 1e{-}05, 1e{-}04\}$ |
| *ReDO* | |
| Dormancy threshold | $\{0, 0.01, 0.1\}$ |
| Recycling period | $\{10, 100, 1000, 10000, 100000\}$ |
| *CBP* | |
| Replacement rate | $\{0.00003, 0.0001, 0.0003, 0.001, 0.003\}$ |
| Maturity threshold | 100 |
| Decay rate | 0.99 |
| *Load balancing* | |
| Target range | $\{5{-}95\%, 20{-}80\%, 40{-}60\%, 50\%\}$ |
| Step size | 0.001 |
| EMA decay | 0.99 |

Table 2: Hyperparameters for class-incremental CIFAR-100.

| Hyperparameter | Value |
| --- | --- |
| Number of tasks | 20 |
| Epochs per task | 200 |
| Early stopping | Yes |
| Batch size | 128 |
| Optimizer | SGD |
| Base learning rate | 0.1 |
| LR schedule (per task) | Step decay at epochs 60, 120, 160 |
| Learning rate decay | 0.2 |
| Momentum | 0.9 |
| *L2 init* strength | $\{0.01, 0.001, 0.0001, 0.00001\}$ |
| *S&P* noise scale | 1e−05 |
| *ReDO* | |
| Dormancy threshold | $\{0, 0.01, 0.1\}$ |
| Recycling period | $\{10, 100, 1000, 10000, 100000\}$ |
| *CBP* | |
| Replacement rate | 1e−05 |
| Maturity threshold | 1000 |
| Decay rate | 0.99 |
| *Load balancing* | |
| Target range | $\{5−95\%, 20−80\%, 40−60\%, 50\%\}$ |
| Step size | 0.001 |
| EMA decay | 0.99 |

# E   ACCURACY TABLES

Tables 3 to 5 report final test accuracies/scores (with mean $\pm$ standard error) for all methods on the three benchmarks.

Table 3: **Continual ImageNet:** Final test accuracies ($\pm$ standard error of the mean), averaged over tasks 1900–2000.

| Method | Accuracy (%) |
|---|---|
| Non-affine LayerNorm + L2 | $90.5 \pm 1.7$ |
| LB 50% | $89.8 \pm 1.8$ |
| GELU + LB 50% | $89.8 \pm 1.8$ |
| LeakyReLU + LB 50% | $89.6 \pm 1.8$ |
| LB 40–60% | $89.5 \pm 1.9$ |
| CBP @ BN + L2 | $89.3 \pm 1.9$ |
| LB 20–80% | $89.1 \pm 1.9$ |
| LB 5–95% | $88.8 \pm 1.9$ |
| CBP @ Baseline | $88.8 \pm 1.9$ |
| BN + L2 | $88.8 \pm 1.9$ |
| ReDO @ BN + L2 | $88.7 \pm 1.9$ |
| S&P + BN | $88.7 \pm 1.9$ |
| GELU | $88.6 \pm 1.9$ |
| Non-affine BN + L2 | $88.6 \pm 1.9$ |
| L2 only | $88.6 \pm 2.0$ |
| BN only | $88.5 \pm 2.0$ |
| S&P | $88.3 \pm 2.0$ |
| LeakyReLU | $88.3 \pm 1.9$ |
| LayerNorm only | $87.2 \pm 2.2$ |
| L2 init | $87.2 \pm 2.1$ |
| LayerNorm + L2 | $87.1 \pm 2.2$ |
| L2 init + BN | $86.3 \pm 2.1$ |
| ReDO @ Baseline | $81.7 \pm 2.4$ |
| Baseline | $79.0 \pm 2.7$ |

Table 4: **Class-incremental CIFAR-100:** Final test accuracies ($\pm$ standard error of the mean), averaged over tasks 15–20.

| Method | Accuracy (%) |
|---|---|
| LB 50% | $78.0 \pm 0.1$ |
| LeakyReLU + LB 50% | $77.9 \pm 0.3$ |
| Normalized skip connection | $77.4 \pm 0.2$ |
| LB 40–60% | $77.1 \pm 0.1$ |
| LB 20–80% | $77.1 \pm 0.1$ |
| LB 5–95% | $77.1 \pm 0.1$ |
| Non-affine BatchNorm | $77.0 \pm 0.1$ |
| L2 init | $76.9 \pm 0.1$ |
| CBP | $76.7 \pm 0.1$ |
| GELU + LB 50% | $76.7 \pm 0.3$ |
| Resets | $76.6 \pm 0.1$ |
| LeakyReLU | $75.9 \pm 0.3$ |
| GELU | $75.7 \pm 0.6$ |
| S&P | $75.5 \pm 0.2$ |
| Baseline | $74.4 \pm 0.2$ |
| ReDO | $72.7 \pm 0.2$ |

Table 5: **Rainbow DQN:** Final human-normalized scores ($\pm$ standard error of the mean) using a time-weighted exponential moving average.

| Environment | Baseline | LayerNorm | LN + LB 50% | CBP |
|---|---|---|---|---|
| BattleZone | $0.7 \pm 0.1$ | $1.0 \pm 0.1$ | $1.0 \pm 0.1$ | $0.8 \pm 0.1$ |
| DemonAttack | $50.6 \pm 3.5$ | $47.8 \pm 4.5$ | $\mathbf{55.9} \pm 1.1$ | $52.1 \pm 2.3$ |
| DoubleDunk | $8.0 \pm 0.1$ | $7.9 \pm 0.2$ | $8.0 \pm 0.3$ | $8.0 \pm 0.2$ |
| NameThisGame | $1.2 \pm 0.2$ | $1.8 \pm 0.2$ | $\mathbf{2.0} \pm 0.1$ | $1.6 \pm 0.2$ |
| Phoenix | $2.5 \pm 0.8$ | $6.3 \pm 1.6$ | $\mathbf{7.4} \pm 1.3$ | $3.6 \pm 0.4$ |
| Qbert | $1.4 \pm 0.1$ | $1.4 \pm 0.1$ | $\mathbf{1.6} \pm 0.1$ | $1.2 \pm 0.1$ |

# F  RELATION TO EXISTING PLASTICITY METRICS

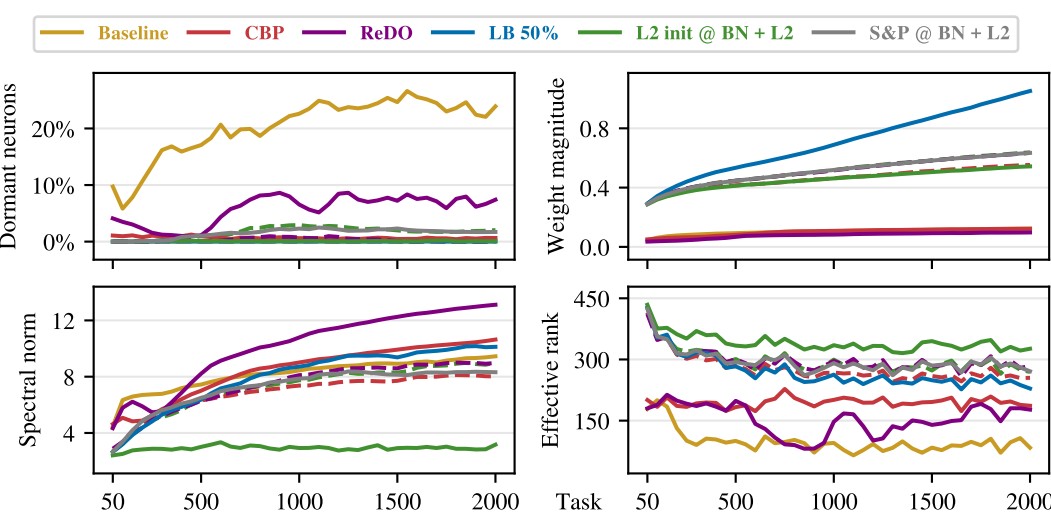

Figure 14: **Continual ImageNet:** Top row: fraction of dormant neurons (left; threshold $\tau = 0.01$) and weight magnitudes (right) over tasks. Bottom row: overall maximum spectral norm across layers and effective rank (Lewandowski et al., 2025). Dashed lines indicate variants that are built on the "BN + L2" configuration.

**Dormant neurons.** A common proxy for plasticity is the fraction of *dormant* neurons (Sokar et al., 2023). Following their definition, a neuron's *score* is its activation magnitude, normalized by the magnitudes of other neurons in the same layer and averaged over the dataset; a neuron is labeled dormant if its score falls below a threshold $\tau$ (we use $\tau = 0.01$). In Figures 14 and 15, all proposed approaches maintain a near-zero fraction of dormant units relative to their respective baselines. An exception is non-affine LayerNorm on continual ImageNet, which shows more dormant units. This is consistent with the firing-rate heatmaps for the same setting, where some layers exhibit occasional 0% firing (see the last row of Figure 18). Overall, 0% firing is a sufficient (though not necessary) condition for dormancy, explaining the observed alignment between the metrics.

In Figures 14 and 15, we observe that the percentage of dormant neurons across all approaches is lower than the baselines and close to zero. The only exception is the non-affine LayerNorm (the pink line in Figure 14). However, this is also reflected in the firing-rate plots of the same experiment, where the firing rate occasionally drops to 0% (most clearly visible in the last row of Figure 18). Dormancy primarily captures the low-activity neurons. Firing-rate distributions are broader, as they reveal low-rate collapse, high-rate linearization, and intermediate-range activity.

**Weight magnitudes.** Plasticity loss is often accompanied by growth in weight norms (Dohare et al., 2024). We observe the same trend in Figures 14 and 15. Methods that avoid collapse also

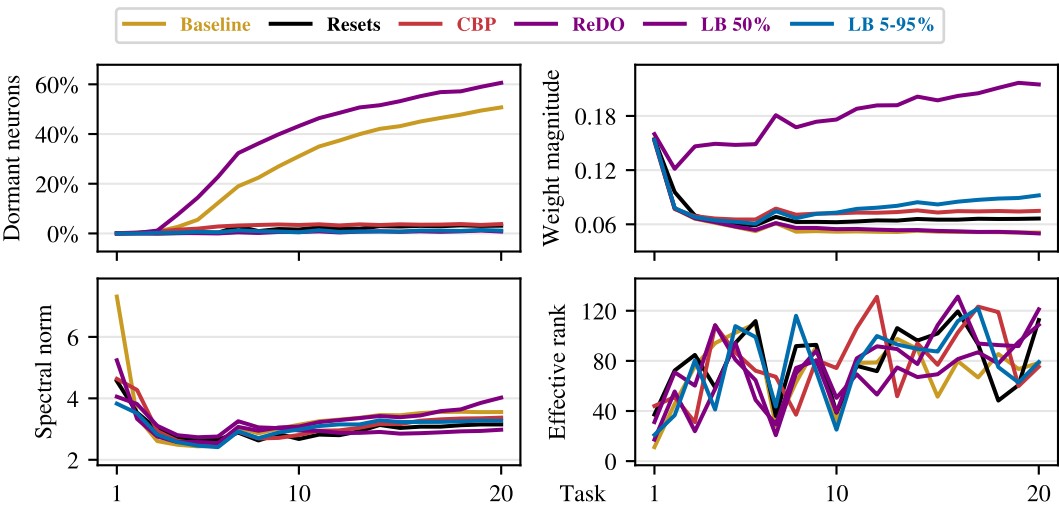

Figure 15: **Class-incremental CIFAR-100:** Top row: fraction of dormant neurons (left; threshold $\tau = 0.01$) and weight magnitudes (right) over tasks. Bottom row: overall maximum spectral norm across layers and effective rank (Lewandowski et al., 2025).

limit the increase in weight magnitudes, with non-affine normalization yielding particularly small magnitudes. A notable exception is the load-balancing setup, where larger biases can appear in the preceding BatchNorm layers (the shift competes with the learned bias).

**Spectral norm and effective rank.** Lewandowski et al. (2025) have shown that the spectral norm of the network layers and the effective rank are related to plasticity. In Figures 14 and 15, we evaluate the overall maximum spectral norm across layers, and the effective rank.

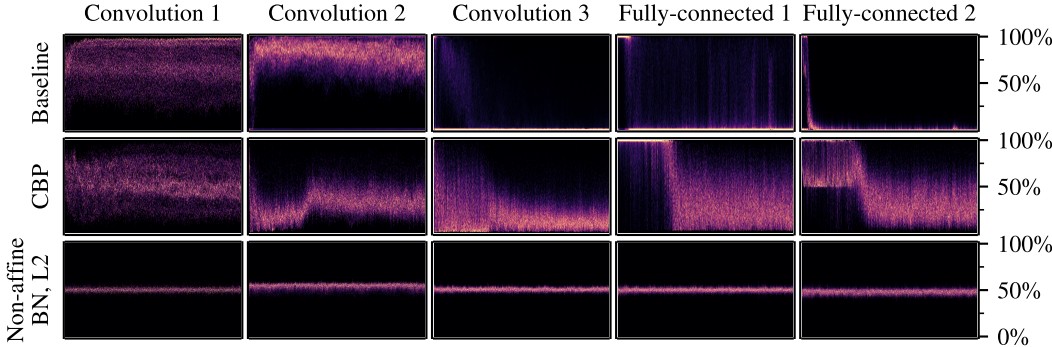

Figure 16: **Continual ImageNet:** Layer-wise firing rate heatmaps over tasks. Rows correspond to methods, columns correspond to successive layers. Early layers stay active while later layers collapse in the baseline; CBP shows heterogeneous behavior; non-affine BN + L2 maintains near-blanced firing rates across layers.

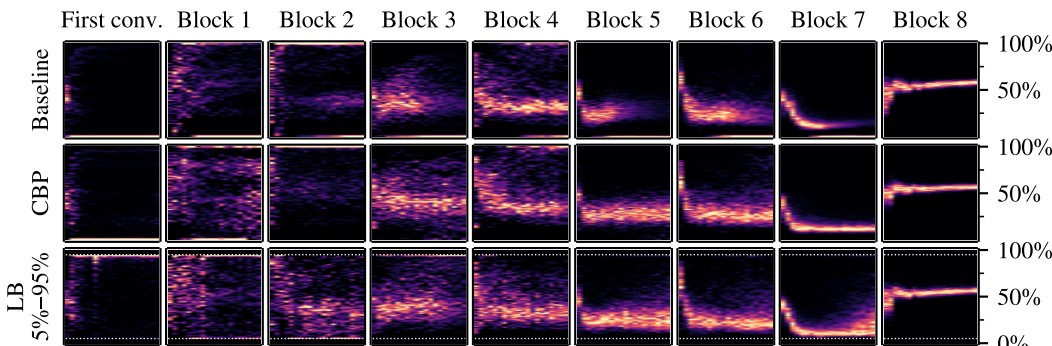

Figure 17: **Class-incremental CIFAR-100:** Layer-wise firing rate heatmaps over tasks. Continual training and CBP exhibit linearization in Block 2.

## G  LAYER-WISE FIRING RATES

Thus far we have visualized firing rates aggregated over all neurons. Layer-wise views reveal where collapse or linearization emerges. All heatmaps follow the visualization convention introduced in the main text.

In Figure 16, we show layer-wise firing-rates dynamics for continual ImageNet. In the first row (baseline), the first two convolutional layers sustain high activity while subsequent layers collapse toward 0%. In the second row (CBP), activity varies considerably across layers over time. By contrast, non-affine BatchNorm + L2 (third row) maintains a stable, near-balanced firing rates across layers. Comprehensive results for all approaches are provided in Figure 18.

In Figure 17 (class-incremental CIFAR-100), we observe linearization emerging in the second ResNet block (Block 2) under continual training and CBP. Load balancing with an upper limit of 95% prevents this effect, which may contribute to its higher test accuracy. However, under load balancing, firing rates sometimes accumulate at the 95% limit in early layers, indicating saturation that is also suboptimal. Comprehensive results for all approaches are provided in Figure 19.

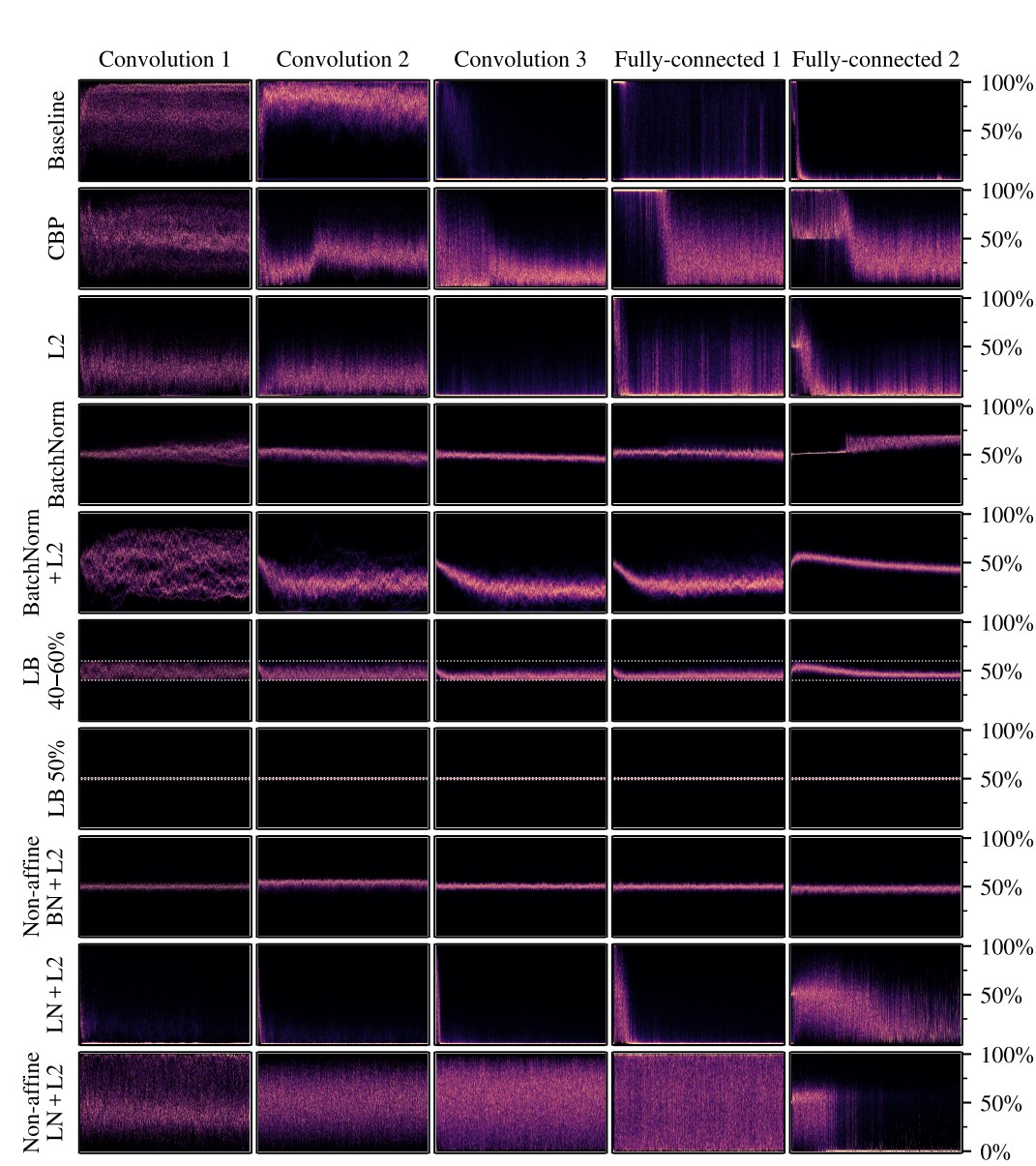

Figure 18: **Continual ImageNet:** Layer-wise firing-rate distrubutions for all approaches.

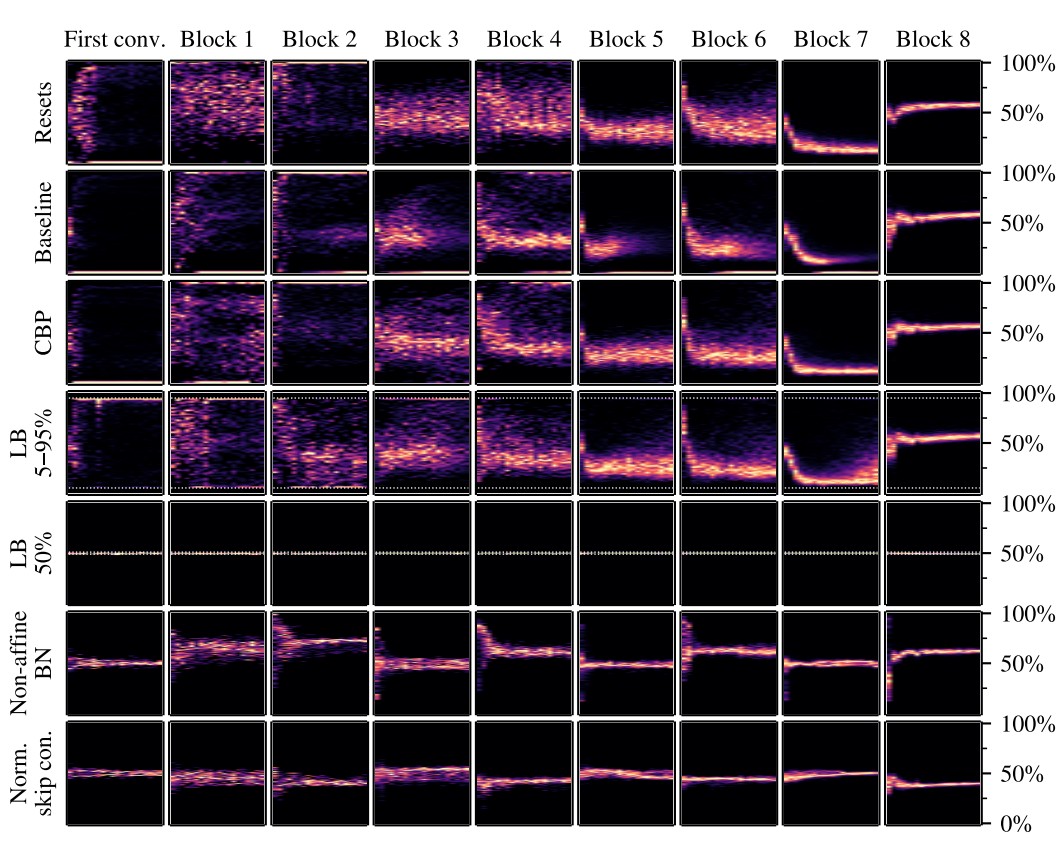

Figure 19: **Class-incremental CIFAR-100:** Layer-wise firing rate distrubutions for all approaches.

# H  ALGORITHM

Load-balancing layer in PyTorch (for 1D inputs, e.g., after fully connected layers with $d$ features):

```python
class LoadBalancer(nn.Module):
  def __init__(self, d, min_rate, max_rate, momentum=0.99, step=0.001):
    super().__init__()
    self.min_rate = min_rate
    self.max_rate = max_rate
    self.momentum = momentum
    self.step = step
    self.register_buffer("avg_rate", 0.5 * torch.ones(d))
    self.register_buffer("offset", torch.zeros(d))

  def forward(self, x):
    x = x + self.offset.unsqueeze(0)

    if self.training:
      with torch.no_grad():
        rate = (x > 0).float().mean(dim=0)
        self.avg_rate *= self.momentum
        self.avg_rate += (1 - self.momentum) * rate
        below_mask = (self.avg_rate < self.min_rate).float()
        above_mask = (self.avg_rate > self.max_rate).float()
        self.offset += self.step * (below_mask - above_mask)

    return x
```

For 3D inputs (after convolutional layers), replace two lines:

```python
x = x + self.offset.reshape(1, -1, 1, 1)
rate = (x > 0).float().mean(dim=[0, 2, 3])
```

**Best practices.**  In our experiments, the EMA decay $\tau = 0.99$ and step size $\alpha = 0.001$ were robust across settings. We recommend starting with a broad target range (e.g., 5-95%) and, if helpful, tightening toward 50% (which we implement as 49-51%). As mentioned in Section 4, we also recommend using load balancing together with a normalization layer placed before it. We always insert the load-balancing layer immediately before the activation function; when a normalization layer is present, this means placing load balancing between the normalization and the activation.

