# OpenReview forum: "Load Balancing Neurons: Controlling Firing Rates Improves Plasticity in Continual Learning"
_ICLR.cc/2026/Conference — Submitted to ICLR 2026_

### Official Review · Reviewer_EQ9A · 2025-10-31

**Soundness:** 3
**Presentation:** 4
**Contribution:** 3
**Rating:** 8
**Confidence:** 4

**Summary:**

In this paper the authors argue that neuron firing rates (fraction of positive pre-activations for ReLU-like units) provide a simple, interpretable lens on plasticity loss in continual learning. They document how standard training drives many units to dead (≈0%) or always-on (≈100%) states, and they correlate with balanced activation with improved adaptation.

Building on this, the authors propose a lightweight load-balancing (LB) layer that adds a per-neuron, stop-gradient offset β before ReLU and updates it with a tiny controller to keep each neuron’s exponentially averaged firing rate ρˉ within target bounds (e.g., 40–60% or exactly 50%).

Across Continual ImageNet (binary class stream) and class-incremental CIFAR-100, LB improves accuracy over strong baselines (BatchNorm+L2, resets, and CBP), and the paper further shows that non-affine normalization and normalized skip connections naturally steer firing rates toward balanced regimes and also help. There are small studies with LeakyReLU/GELU and an RL demo on DemonAttack (DQN) suggesting similar trends. Code snippets and detailed training protocols are provided in the appendix.

**Strengths:**

- Clear and interpretable diagnostic. Firing-rate distributions over tasks/layers reveal dead/linearized regimes and track plasticity dynamics better than scalar “dormancy” alone. The layer-wise heatmaps are especially informative.

- Simple, low-overhead mechanism. The LB layer (stop-gradient offset with EMA-based controller) is easy to drop-in, architecture-agnostic, and empirically effective with tiny step sizes and bounds.

- Consistent gains across setups. On Continual ImageNet and CIFAR-100 CIL, LB (40–60% or 50%) improves over BN+L2, resets, and often CBP. The tables summarize statistically stable gains.

- Architectural guidance. The authors infer that under non-affine normalization and normalized skip connections push firing rates toward balanced regimes and can match or exceed CBP, offering alternative routes when LB is undesirable.

- Transparency & scope. Training details, implementation notes, sensitivity to activations, and an RL probe broaden credibility. The LLM usage is explicitly disclosed.

**Weaknesses:**

- The theoretical justification remains local. The 50% target is motivated by per-neuron entropy; the paper acknowledges that joint entropy/correlation matters but does not measure it. Some empirical cases (non-affine LayerNorm) suggest layer-dependent optima instead of a universal 50%. A small study of inter-neuron correlations / total correlation would strengthen the story.

- Bound & sensitivity of the controller. The LB controller uses fixed α,τ,[ρmin⁡,ρmax⁡]. There is limited analysis of stability/sensitivity to these hyperparameters across tasks/architectures, class imbalance, or nonstationarity patterns.

- Broader CL baselines. The comparisons focus on CBP, resets, BN/LN(+L2). Missing are rehearsal-based and regularization-based class-incremental baselines (e.g., ER-ACE/DER++, EWC/SI, LwF/distillation, prompt/adapters), which might help to contextualize practical competitiveness.

- Potential interaction with biases/norms. Figures suggest weight/bias magnitudes shift under LB (BN bias competition). A brief calibration/regularization analysis (e.g., weight decay placement) might be the way that can help practitioners.

**Questions:**

- Controller robustness. How sensitive are gains to α, τ, and the band [ρmin⁡,ρmax⁡]? Would be nice if the authors provide a grid on CIFAR-100 CIL and Continual ImageNet, including tighter/looser bands and varied EMA half-lives.

- When not to use 50%? Non-affine LN appears to benefit from broader, layer-varying firing-rate distributions, and enforcing 50% might lead to reduced accuracy. Would a layer-adaptive target (e.g., via percentiles or learned priors) be valuable, can the authors elaborate/explain on the benefits (or report results)?

- Correlations across units. Would be nice to see total correlation / redundancy among activation states to complement the marginal-entropy view and test whether LB reduces harmful correlations.

- Compatibility with rehearsal / adapters. How does LB interact with experience replay (balanced buffers) or adapter/prompt methods?

- Conv vs. FC placement. The authors show 1D/conv variants. Any cases where inserting LB after vs. before normalization helps/hurts? The work might benefit from clarifying the recommended placement when norms are absent or affine.

**Details Of Ethics Concerns:**

N/A.

---

> ### Author Response · Authors · 2025-11-25
>
> We thank the reviewer for the careful reading and constructive feedback, and for the concrete suggestions. Below we respond to the main points and cite the corresponding additions in the revision.
>
> > The theoretical justification remains local [...]
>
> Agreed. In the revision we (i) updated the theory to motivate plasticity at the level of activation patterns / codes (not just single neurons), and (ii) added an empirical analysis of marginal entropies and total correlation (new Fig. 5). This directly checks the missing piece: even though LB can increase redundancy somewhat, the marginal entropies increase more, so overall code diversity increases.
>
> > Bound & sensitivity of the controller. The LB controller uses fixed α,τ,[ρmin⁡,ρmax⁡]. There is limited analysis of stability/sensitivity to these hyperparameters across tasks/architectures, class imbalance, or nonstationarity patterns.
>
> We agree. In practice, we found $\alpha = 0.001$ and $\tau = 0.99$ to be stable across our main settings, and we now state recommended defaults and placement more clearly (new "Best practices" paragraph in Appendix D). For the band, we recommend starting wide (5-95%) to prevent the common failure modes (near-0% death and near-100% saturation), and only tightening toward 49-51% if needed. Strict 50% is a stronger intervention and can change training dynamics, so we do not treat it as a universal default.
> We plan to add a larger sweep where possible, but we did not complete a full sensitivity grid within this revision.
>
> > Broader CL baselines. The comparisons focus on CBP, resets, BN/LN(+L2). Missing are rehearsal-based and regularization-based class-incremental baselines (e.g., ER-ACE/DER++, EWC/SI, LwF/distillation, prompt/adapters), which might help to contextualize practical competitiveness.
>
> We agree broader baselines would be useful for context. In this revision we prioritized strengthening the empirical and analytical evidence for our claims. As mentioned under "Ongoing experiments", we are currently evaluating L2-init and adding additional plasticity indicators (spectral norm and gradient-covariance). We will include additional baselines where feasible in the next revision.
>
> > Potential interaction with biases/norms. Figures suggest weight/bias magnitudes shift under LB (BN bias competition). A brief calibration/regularization analysis (e.g., weight decay placement) might be the way that can help practitioners.
>
> Our understanding is that this refers to a practical effect: the LB shift $\beta$ can "fight" with the learned bias in the layer before it (the model can partly cancel the shift by adjusting biases). This is why we recommend using LB together with a normalization layer placed before it. We now make these recommendations explicit.
> Could you confirm that this is the interaction you had in mind (and that these clarifications address the point), or were you referring to a different calibration issue?
>
> > Controller robustness. How sensitive are gains to α, τ, and the band [ρmin⁡,ρmax⁡]? Would be nice if the authors provide a grid on CIFAR-100 CIL and Continual ImageNet, including tighter/looser bands and varied EMA half-lives.
>
> In our experiments, the controller parameters worked well across the main settings without requiring careful tuning. The target band is most influential: a wide band (5–95%) typically prevents collapse, while tighter bands near 50% are a stronger intervention and can help or hurt depending on the setting. We did not complete a full grid over class-incremental CIFAR-100 and Continual ImageNet in this revision, but plan to add it where feasible.
>
> > Correlations across units. Would be nice to see total correlation / redundancy among activation states to complement the marginal-entropy view and test whether LB reduces harmful correlations.
>
> Addressed via the new total correlation analysis (new Fig. 5), which complements the marginal-entropy view.
>
> > Compatibility with rehearsal / adapters. How does LB interact with experience replay (balanced buffers) or adapter/prompt methods?
>
> Our updated RL runs use Rainbow, which includes experience replay, and LB works without issues there. We did not test adapter/prompt methods yet; we agree this is interesting future work.
>
> > Conv vs. FC placement. The authors show 1D/conv variants. Any cases where inserting LB after vs. before normalization helps/hurts? The work might benefit from clarifying the recommended placement when norms are absent or affine.
>
> Our recommendation is consistent across conv/FC layers: place LB immediately before the activation to directly control the pre-activations whose sign defines the gate. When a normalization layer is present, place LB between normalization and activation (norm, LB, activation). When norms are absent, LB still functions, but we recommend pairing it with normalization when possible (as noted in Section 4) to reduce bias competition and stabilize calibration.

---

### Official Review · Reviewer_UUtk · 2025-10-31

**Soundness:** 2
**Presentation:** 3
**Contribution:** 2
**Rating:** 4
**Confidence:** 4

**Summary:**

This paper analyses plasticity loss from the perspective of neuron firing rates. The authors posit that a neuron carries maximum information when the entropy of the indicator of firing is maximized; which is precisely at a firing rate of 50%. The authors introduce the load balancing mechanism that biases each neuron towards a range of target firing rates. Training a ResNet-18 on the Continual ImageNet and Class Incremental CIFAR 100 continual learning benchmark problems, the authors show that firing rates that deviate away from 50% are a correlate of plasticity loss and that their proposed load balancing method maintains the target firing rate while improving plasticity over competitor continual learning interventions. Additionally, the authors briefly analyze normalization schemes, remarking that the success of these methods can be tied to their ability to perform “load balancing”.

**Strengths:**

- The paper is well written and clear.
- The manner in which the authors analyze plasticity loss through firing rates is simple and avoids unnecessary abstractions. Motivating the optimal firing rate of 50% through the lens of maximizing the information of a neuron is intuitive and simple. Connecting firing rates to neuron death in conjunction with teh analysis of normalization schemes under this framework is a positive unification of existing work.
- The presentation of the experimental results is relatively clear and supports the claims of the paper.

**Weaknesses:**

- The paper claims to introduce neuron firing rates to the analysis of plasticity loss. However, there is existing work [2] that casts the problem of identifying dead neurons as that of identifying neurons whose firing rates have effectively halted. It would be useful to cite this work and juxtapose its analysis and the Self Normalized Resets (SNR) algorithm with that of the authors’ analysis and the load balancing method.
- While the lens of firing rates is interesting, it is not entirely novel (see the point above). For instance, it is not clear if a decline in the firing rate is not simply another way of describing neuron death. It would be useful in the main body to showcase when deviations from the optimal firing rate are synonymous with neuron death and when they are not. In fact, almost all of the plots with side-by-side firing rates and loss/accuracy, show that plasticity loss is associated with a collapse in the firing rate, which is essentially neuron death. However, the authors make the point that a suboptimal firing rate could be above 50%, but none of the empirical evidence demonstrates that this occurs.
- Crucially, the claims of the paper would be strengthened if additional competitor algorithms were evaluated against load balancing. For instance, only a single continual reset scheme, CBP, is evaluated, but ReDO [1] and SNR [2] have not been evaluated. Given how tangential SNR is to load balancing and its robust performance, it would be useful to consider this method. Additionally, L2 Init [3] would also be critical to evaluate as this is a simple, robust, and effective regularization based approach for mitigating plasticity loss. It would be worthwhile to investigate neuron firing rates under L2 Init.
- The paper considers a limited number of benchmark problems: only Continual ImageNet and Class Incremental CIFAR 100. Evaluating load balancing on more benchmarks would strengthen the results.
- The paper considers only a CNN and a ResNet-18 architecture. Additional architectures in the experiments would be useful, such as a ViT.
- It would be useful to have a section (say in the appendix) listing the hyperparameters used in the experiments and how they were selected. Were competitor algorithms tuned fairly against load balancing, beyond simply using the same hyperparameters from prior literature?
- The empirical results show a minor improvement in loss/accuracy with load balancing over competitor methods. Given how minor the improvement is, along with limited set of competitor methods, architectures, and benchmark problems evaluated along with a lack of details of hyperparameter tuning, the efficacy of load balancing is in question.


[1] Sokar, Ghada, et al. "The dormant neuron phenomenon in deep reinforcement learning." International Conference on Machine Learning. PMLR, 2023.
[2] Farias, Vivek F., and Adam D. Jozefiak. "Self-normalized resets for plasticity in continual learning." arXiv preprint arXiv:2410.20098 (2024).
[3] Kumar, Saurabh, Henrik Marklund, and Benjamin Van Roy. "Maintaining plasticity in continual learning via regenerative regularization." arXiv preprint arXiv:2308.11958 (2023).

**Questions:**

Related to the issue of hyperparameters, do the experiments use SGD or Adam and with what learning rate?

---

> ### Author Response · Authors · 2025-11-25
>
> We thank the reviewer for the careful reading and constructive feedback. Below we respond point-by-point and reference the relevant changes in the revised manuscript.
>
> > The paper claims to introduce neuron firing rates to the analysis of plasticity loss [...]
>
> Agreed. As noted in the Official Comment (Relation to SNR), we added Self-Normalized Resets (SNR) to the background and explicitly contrast it with load balancing (both use per-neuron firing behavior online; SNR is reset-based via inter-firing-time tail tests, ours is continuous threshold control without reinitialization).
>
> > While the lens of firing rates is interesting, it is not entirely novel [...]
>
> We address this in the Official Comment (Firing-rate deviations are not synonymous with dead ReLUs). Briefly: "dead" corresponds to $p \approx 0$, but firing-rate diagnostics also expose the opposite extreme $p \approx 1$ (saturation/linearization: gradients flow but gating becomes nearly constant, reducing activation-pattern diversity).
> We show saturation/linearization behavior in CBP dynamics (Fig. 1), GELU (Fig. 6), and Rainbow RL (Fig. 12), and also show that strong performance can coincide with spread-out firing rates (non-affine LayerNorm, Fig. 9). This is why we emphasize layer-wise firing-rate distributions rather than a binary dead/alive criterion.
>
> > Crucially, the claims of the paper would be strengthened if additional competitor algorithms were evaluated against load balancing [...]
>
> We agree this would strengthen the evaluation. As noted in the "Ongoing experiments", we are currently evaluating L2-init and adding complementary plasticity indicators (spectral norms, gradient-covariance).
> We will incorporate additional baselines where feasible.
>
> > The paper considers a limited number of benchmark problems: only Continual ImageNet and Class Incremental CIFAR 100. Evaluating load balancing on more benchmarks would strengthen the results.
>
> We agree. In this revision we expanded the RL evaluation by updating to Rainbow DQN on 6 games (Appendix B.1 / Section 5), to complement Continual ImageNet and class-incremental CIFAR-100.
>
> > The paper considers only a CNN and a ResNet-18 architecture. Additional architectures in the experiments would be useful, such as a ViT.
>
> We agree it's interesting, but we consider this outside the scope of the current paper: our contribution is primarily a mechanism + diagnostic lens for ReLU-style gating dynamics, and our experiments focus on widely used continual-learning CNN/ResNet settings.
>
> > It would be useful to have a section (say in the appendix) listing the hyperparameters used in the experiments and how they were selected. Were competitor algorithms tuned fairly against load balancing, beyond simply using the same hyperparameters from prior literature?
>
> We expanded Appendix D with hyperparameter tables and a clearer description of what was tuned and over what ranges.
>
> > The empirical results show a minor improvement in loss/accuracy with load balancing over competitor methods. Given how minor the improvement is, along with limited set of competitor methods, architectures, and benchmark problems evaluated along with a lack of details of hyperparameter tuning, the efficacy of load balancing is in question.
>
> We understand the concern. In continual learning, even small average accuracy gains can reflect noticeably better retention over many tasks, and in our results the gains consistently align with firing-rate dynamics (collapse vs. maintained utilization). Load balancing is also lightweight and non-destructive (no reinitialization), so it provides a simple, architecture-agnostic way to stabilize plasticity without discarding learned information. Finally, the new entropy/total-correlation analysis (Fig. 5) supports the mechanism beyond accuracy alone.
>
> > Related to the issue of hyperparameters, do the experiments use SGD or Adam and with what learning rate?
>
> We use SGD for Continual ImageNet and class-incremental CIFAR-100, and Adam for the new Rainbow RL experiments; full learning-rate details and schedules are listed in Appendix D.

---

### Official Review · Reviewer_nCj4 · 2025-11-01

**Soundness:** 2
**Presentation:** 3
**Contribution:** 2
**Rating:** 2
**Confidence:** 4

**Summary:**

## Summary
* The paper proposes a firing-rate analysis based on neuron activation probabilities and demonstrates that it can serve as a lightweight tool to diagnose plasticity loss.
* It further introduces a load-balancing method that mitigates plasticity loss by explicitly controlling firing rates.
* Through continual learning and reinforcement learning experiments, the paper shows the effectiveness of the proposed approach.

## Pros
* Proposes a simple and intuitive new diagnostic tool for plasticity loss.
* Conducts experiments and analyses on established benchmarks for assessing plasticity loss, including continual ImageNet, class-incremental CIFAR-100, and a reinforcement learning setting.

## Cons
* Insufficient analysis of the proposed load-balancing method itself.
* Experimental scope is limited and focuses on a specific architecture.
* Some details are missing from the main text.

**Strengths:**

- (originality) Proposes firing rate as a new metric for measuring plasticity. This metric unifies two known phenomena arising from plasticity loss—dormant neurons and linearized neurons—and is highly intuitive.
- (quality) The paper uses appropriate baselines in experiments. In addition to methods proposed by prior work, it also considers common training variants such as BN+L2, clarifying the interpretation of results.
- (clarity) Both the proposed metric and methodology are clearly described; the experimental settings and results are easy to understand.
- (significance) Beyond the specific method, the paper explores architectural design choices that can naturally induce balanced firing rates, suggesting broad applicability.

**Weaknesses:**

- The proposed methodology relies on adding a normalization layer, which is a limitation; the paper does not discuss the additional computational cost.
- [Sec 3.1] The claim that the optimal firing rate is 50% lacks theoretical justification. Is there evidence that maximizing mutual information leads to improved plasticity? If we interpret the firing rate as an activation probability, doesn’t enforcing it risk constraining model expressivity?
- [Sec 5] The experimental scope is too limited. In particular, the RL study uses only a single task and does not include comparisons with methods such as CBP.
- [Sec 5.2] The reinforcement learning experiment lacks results and analysis in the main text; at minimum, a brief discussion is needed.
- The paper omits analyses of several recently discussed plasticity-related metrics. Including additional indicators such as spectral norm and gradient covariance would strengthen the study [R1].
[R1] Alex Lewandowski et al., Learning Continually by Spectral Regularization, ICLR 2025.

**Questions:**

- [L296] What is the rationale for choosing $\alpha = 0.001$, and how does the actual $\rho$ value change after the shift?
- [L302] After applying Load Balancing and BatchNorm, isn’t the resulting functional change too large? Although CBP is described as a reset-based method, it induces little functional change in practice.
- [L302] What is the reason for applying L2?
- [Figure 3–4] What happens if BatchNorm and L2 are also applied to CBP? Do we obtain essentially the same performance as simply applying L2 to CBP?
- The defined neuron activity appears to reflect each neuron’s contribution to learning, but a neuron with low activity may still encode important information. Have you examined weight utility or related measures?
- [Sec 4] Could the optimal firing rate be neuron-specific? Farias and Jozefiak [R2] highlight that activation probabilities can vary across neurons.

[R2] Vivek Farias and Adam Daniel Jozefiak, Self-Normalized Resets for Plasticity in Continual Learning, ICLR 2025.

---

> ### Author Response · Authors · 2025-11-25
>
> We thank the reviewer for the careful reading and constructive feedback. Below we address the concerns raised and point to where each is handled in the revised manuscript.
>
> > The proposed methodology relies on adding a normalization layer, which is a limitation [...]
>
> We agree this is important to clarify. In several of our settings normalization is already part of the standard backbone (e.g., class-incremental CIFAR-100 uses normalized ResNet blocks), so load balancing does not introduce an additional normalization layer there. For continual ImageNet we add BatchNorm, and for RL we add LayerNorm, since we found a preceding normalization helpful for stability (to avoid the learned bias competing with the LB shift). The computational overhead of load balancing itself is small: it maintains two scalars per neuron (the shift $\beta$ and an EMA firing rate) and applies a constant shift plus a few lightweight operations per activation. Empirically on continual ImageNet, our runs take $\approx$ 2h20m with BatchNorm and $\approx$ 2h40m with BatchNorm + load balancing.
>
> > The claim that the optimal firing rate is 50% lacks theoretical justification [...]
>
> Addressed in the revision (see Official Comment: Improved theory). We now motivate firing rates via activation-pattern entropy / effective region usage, and add a total-correlation analysis (new Fig. 5) to move beyond a purely per-neuron ("local") argument. We also clarify the intended claim: pushing gates toward intermediate regimes is a simple pressure against collapse; we do not claim that forcing every unit to exactly 50% is universally optimal for accuracy.
>
> > The experimental scope is too limited [...]
>
> We switched the RL setup to Rainbow DQN and expanded to multiple Atari games (Appendix B.1 / results table). We are currently running CBP as an additional RL baseline and will include it in the rebuttal if it finishes before the deadline.
>
> > The reinforcement learning experiment lacks results and analysis in the main text [...]
>
> We added the full RL setup + curves + firing-rate plots in Appendix B.1 and a results table (Appendix F).
>
> > The paper omits analyses of several recently discussed plasticity-related metrics [...]
>
> Agreed. As noted in "Ongoing experiments", we plan to add spectral norm and gradient-covariance in the next revision.
>
> > [L296] What is the rationale for choosing $\alpha = 0.001$, and how does the actual $\rho$ value change after the shift?
>
> We chose $\alpha$ as a small step that behaved stably in preliminary sweeps. An update of $\beta$ changes the firing probability as $\rho = \Pr(x + \beta \geq 0) = \Pr(x \geq -\beta)$,
> so increasing $\beta$ increases $\rho$.
> In practice the EMA smoothing means the average firing rate changes gradually, which makes the controller stable.
>
> > [L302] After applying Load Balancing and BatchNorm, isn’t the resulting functional change too large? [...]
>
> We understand the concern. For CIFAR-100, CBP already uses BatchNorm in our setup, so the comparison is on equal architectural footing. For Continual ImageNet, we agree this fairness point matters; we are currently running a CBP + BatchNorm variant and will include the results in the rebuttal if they finish before the deadline.
>
> > [L302] What is the reason for applying L2?
>
> Two reasons: (i) it helps prevent $\beta$ from "fighting" with upstream biases/scale drift by keeping weights/biases from growing unnecessarily, and (ii) it lets us evaluate load balancing on top of a strong baseline (BN+L2) so any gain is not just "benefit of adding regularization."
>
> > [Figure 3–4] What happens if BatchNorm and L2 are also applied to CBP? [...]
>
> This is a good suggestion. As mentioned above, we are running BatchNorm on top of CBP and will include the results in the rebuttal if they finish before the deadline.
>
> > The defined neuron activity appears to reflect each neuron's contribution to learning, but a neuron with low activity may still encode important information. [...]
>
> We agree that low activity does not necessarily imply low importance: a neuron can encode a useful sparse feature while firing rarely. Our firing-rate analysis is intended as a distribution-level diagnostic (detecting widespread collapse/saturation), not as a claim that any single low firing-rate neuron is unimportant. In Appendix E we analyze related signals (e.g., weight magnitudes and dormant-neuron indicators), and we are working on adding additional plasticity metrics such as spectral norm and gradient-covariance before the deadline.
>
> > Could the optimal firing rate be neuron-specific? [...]
>
> Yes for accuracy it can be neuron-/layer-dependent (e.g., the non-affine LayerNorm case shows strong performance with broad firing-rate distributions). Our revised framing treats 50% as an entropy-maximizing pressure against collapse, not a universal per-neuron optimum for accuracy; this is also why we recommend starting with a wide band (5-95%) and tightening only if helpful.

---

### Official Review · Reviewer_vFaN · 2025-11-02

**Soundness:** 2
**Presentation:** 3
**Contribution:** 2
**Rating:** 4
**Confidence:** 4

**Summary:**

The paper reframes plasticity loss as an imbalance in neuron activity, units either drift toward dead or always-on regimes. The authors argue that the firing rate (fraction of positive pre-activations) is a simple, layer-agnostic diagnostic to track and prevent this collapse. It motivates an information-theoretic target of 50% firing, maximizing the entropy of the binary activation of each ReLU. Building on this lens, the authors propose a lightweight load-balancing layer that adds a per-neuron, stop-gradient offset before the ReLU and updates it up/down after each batch so the long-term firing rate stays within chosen bounds (e.g., 40–60% or exactly 50%). On Continual and class-incremental CIFAR-100, constraining firing rates to intermediate ranges improves accuracy over baseline methods. The analysis further connects architectural choices to the same principle: non-affine normalization and normalized skip connections implicitly center or stabilize pre-activations, producing more balanced firing distributions.

**Strengths:**

- Identifying neuron firing rates as being a correlate of plasticity is somewhat novel, albeit, tightly correlated with the well documented phenomenon of neuron death.
- The connection between neuron death and normalization schemes through the lens of neuron firing rates is a strength this paper, as this unifies two strategies for mitigating plasticity loss: neuron resets and normalization.
- The main body is clear and avoids unnecessary jargon and abstractions. For instance, the intuition regarding targeting an optimal firing rate of 50% is very approachable.

**Weaknesses:**

- It would be nice if some theory could be derived regarding the choice of targeting a firing rate of 50%. While the information theoretic argument provides good intuition, it is not clear that this is optimal for all architectures, model sizes, and environments.
- To strengthen the claims of the paper, the authors should consider expanding their experiments: more continual learning benchmark problems, more architectures, such as a transformer architecture, and comparing against more competitor algorithms, such as additional reset schemes and regularization schemes.
- There is a recent work on Self Normalized Resets that similarly analyzes the firing rates of neurons and derives a reset scheme based on such information. It would be useful if the authors could contrast their analysis and approach against this method.
- One critique of the overall message: it is not clear that deviating from an optimal firing rate is distinct from neuron death. In addition, most of the experiments seem to illustrate a decline in neuron firing rates when plasticity loss is present.

**Questions:**

None

---

> ### Author Response · Authors · 2025-11-25
>
> We thank the reviewer for the careful reading and constructive feedback. Below we address the points raised and reference the corresponding updates in the revised manuscript.
>
> > It would be nice if some theory could be derived regarding the choice of targeting a firing rate of 50%. While the information theoretic argument provides good intuition, it is not clear that this is optimal for all architectures, model sizes, and environments.
>
> We agree the “50%” message needs to be interpreted correctly. As noted in the official comment (Improved theory), we strengthened Section 3: the revised motivation is network-level (activation-code entropy / effective region usage), with $\sum_i H(S_i)$ used as a tractable proxy and a new entropy/total-correlation analysis (new Fig. 5) supporting that it increases code diversity without excessive redundancy.
> We clarify that $p_i=0.5$ is an entropy-maximizing pressure against collapse, not a claim that it is universally optimal for accuracy across all architectures/environments.
>
> > To strengthen the claims of the paper, the authors should consider expanding their experiments: more continual learning benchmark problems, more architectures, such as a transformer architecture, and comparing against more competitor algorithms, such as additional reset schemes and regularization schemes.
>
> We agree. Within this revision we focused on (i) updating RL to Rainbow DQN and reporting the new results (Appendix B.1 / Section 5), and (ii) adding additional analysis tied to the updated theory (new Fig. 5). As mentioned in the official comment (Ongoing experiments), we are currently evaluating additional baselines (incl. L2-init) and adding spectral norm / gradient-covariance metrics, to be included in the next revision.
>
> > There is a recent work on Self Normalized Resets that similarly analyzes the firing rates of neurons and derives a reset scheme based on such information. It would be useful if the authors could contrast their analysis and approach against this method.
>
> Agreed and addressed. We added SNR to the background and provide a direct comparison in the official comment (Relation to SNR): both are lightweight per-neuron mechanisms using firing behavior online, but SNR is reset-based, whereas our load balancing controls thresholds continually and prevents both dead and saturated regimes.
>
> > One critique of the overall message: it is not clear that deviating from an optimal firing rate is distinct from neuron death. In addition, most of the experiments seem to illustrate a decline in neuron firing rates when plasticity loss is present.
>
> We address this in the official comment (Distinction from dead ReLUs). Briefly: death corresponds to $p \approx 0$, but our diagnostics also capture the opposite extreme $p \approx 1$ (saturation/linearization), which we observe in CBP (Fig. 1), GELU (Fig. 6), and Rainbow RL (Fig. 12).
> Fig. 9 further shows that strong performance can coincide with spread-out firing rates.

---

### Author Response · Authors · 2025-11-25

We thank the reviewers for their thoughtful feedback.
In the revised manuscript, we made several changes to clarify the theory, strengthen the empirical support, and improve the experimental reporting.
Below we give a concise changelog, then summarize the updated theory, our relation to SNR, and why firing-rate deviations are not synonymous with dead ReLUs, and finally note ongoing experiments.

**Changelog**
- Theory (Section 3) clarified/strengthened while preserving the overall narrative. The revised presentation provides a more principled motivation for maximizing marginal entropies (i.e., controlling firing rates) as a proxy objective. We also expanded the complementary theoretical discussion in Appendix C. We removed the previous Appendix B: "Locality" is now integrated into Section 3, and "Magnitude invariance" is no longer needed since the revised theory does not rely on activation magnitudes.
- Added an empirical analysis of activation-pattern entropy and total correlation (Section 4, Figure 5).
- Replaced the outdated DQN with Rainbow DQN and expanded to 6 Atari games. Updated setup, performance curves, and firing-rate plots in Appendix B.1 and a new results table in Appendix F.
- Added Self-Normalized Resets (SNR) to the background section.
- Expanded new Appendix D: training details, hyperparameter tables, and tested ranges.
- Added a short "Best practices" paragraph (default LB hyperparameters and placement).
- Moved "Normalized skip connections" plot to the appendix (now Figure 10) due to page constraints and reorganized appendix sections for readability.

**Improved theory (brief summary)**
Previously, our theoretical motivation was *neuron-local*: a ReLU gate ($S = [x \geq 0]$) has entropy maximized at $p = 0.5$, making firing rates a local diagnostic of dead ($p \approx 0$) or saturated/linearized ($p \approx 1$) units.
The revision makes the argument network-/function-level by focusing on the activation pattern $S(x)\in{0,1}^M$ and data-dependent effective capacity via perplexity $2^{H(S)}$. Since optimizing $H(S)$ directly is difficult, we use $H(S)=\sum_i H(S_i)-TC(S)$,
so maximizing $\sum_i H(S_i)$ is a tractable proxy (up to redundancy (TC)). We then verify empirically that this proxy increases marginal entropies while redundancy remains controlled (new Fig. 5).

**Relation to Self-Normalized Resets (SNR)**
Self-Normalized Resets (Farias & Jozefiak, ICLR 2025) is close in spirit: both are lightweight, per-neuron mechanisms using firing behavior online.
SNR is reset-based: it monitors per-neuron inactivity via inter-firing times and reinitializes a neuron when its time-since-last-activation becomes an extreme tail event under that neuron's own historical firing statistics (a self-normalized percentile test).
In contrast, our load balancing is not a reset scheme: we introduce a continuous feedback controller that shifts each neuron's effective threshold to keep its long-run firing rate within target bounds, thereby preventing both dead and saturated regimes.
Our work emphasizes layer-wise and network-wide firing-rate distributions as a diagnostic signal, which reveals utilization collapse and training anomalies beyond what a single per-neuron statistic captures.

**Distinction from dead ReLUs**
Deviating from "optimal" firing rates is not just a restatement of dead ReLUs.
Neuron death is the specific extreme $p \approx 0$, but firing-rate diagnostics also capture the *other* failure mode $p \approx 1$: units that are almost always on, so gradients still flow yet the gating becomes nearly constant and the network locally linearizes, reducing the diversity of activation patterns.
We see this saturation/linearization regime clearly in the early CBP dynamics (Figure 1), in the GELU setting (Figure 6), and in the new Rainbow RL experiments (Figure 12).
More generally, our contribution is the distributional view: layer-wise firing-rate histograms distinguish (i) collapse to (0%) (inactivity/death), (ii) saturation near (100%) (linearization), and (iii) healthy heterogeneous utilization, e.g., the very spread-out LayerNorm firing rates in Figure 9, which coincide with strong performance.

**Ongoing experiments and forthcoming updates**
In parallel with these revisions, we are continuing to run additional experiments requested by the reviewers.
In particular, we are currently evaluating the L2-init baseline, and we are adding complementary plasticity indicators (spectral norm and gradient covariance).
These results are still in progress and therefore not yet included in the current revision or fully reflected in our point-by-point responses.
We plan to include and discuss the new results in the next revision, which will allow us to address the remaining concerns more comprehensively.

---

> ### Author Response · Authors · 2025-12-03
>
> In our second revision, we substantially expanded and strengthened the experimental evaluation.
>
> **Changelog**
> - We added new baselines as requested by the reviewers, reporting both firing-rate distributions and accuracy. For continual ImageNet and class-incremental CIFAR-100 we include *Shrink-and-Perturb*, *L2 init*, and *ReDO*; for Rainbow we include CBP.
> - We expanded the hyperparameter section, and now report the full sweep ranges and selected settings for all methods.
> - We added requested additional metrics *spectral norm* and *effective rank* (as suggested by reviewer nCj4).

---

### Meta-Review · Area_Chair_UkAH · 2026-01-05

**Summary:**

The paper discusses loss of plasticity from the perspective of both dead neurons and saturated neurons, arguing, with an information-theoretic argument, that one should aspire to have units activating 50% of the time.  The paper then proposes a new layer with the objective of keeping such a balance. The reviewers have praised the paper presentation and the simplicity of the idea, however many concerns were raised about the extent to which the idea has been empirically validated. Reviewers have asked for additional architectures, datasets, and baselines. While the authors did add new results, I struggled to understand them. That might be a consequence of such big changes being made during the rebuttal period, clearly indicating the paper was not ready when first submitted. Let me elaborate: While Figures 1 and 2 depict the firing rate induced by many baselines, I don't see the performance of the same baselines in Figures 3 and 4. Specifically, is S&P or ReDO depicted in those figures? I think not. The same is true for Figure 6. It might be simply that the paper presentation is not super polished, but that would still be an issue by itself.

I do not think the paper should be rejected because it didn't compare itself against every possible baseline in every possible dataset and architecture, but as much as I would gladly accept the paper without additional tests in different architectures or even datasets, I feel that what seems to be incomplete baseline evaluations make acceptance harder, and thus I am recommending this paper to be rejected. I want to encourage the authors to submit the paper with the additional results in a more polished format to the next conference. Additionally, I would encourage the authors to not call their motivation a theoretical result, as that sets the wrong expectation for the readers.

**Reviewer Concerns:**

- _Lack of theoretical results._

	The discussion further clarifies/motivates the proposed idea but it indeed seem to lack a theoretical justification.

- _Not enough empirical results (additional neural network architectures, baselines, and datasets)._

	As expected, the authors have now added additional experiments to the paper.

- _The idea proposed can be seen as derivative of the idea of minimizing neuron death._

	There's more nuance to this claim, such as looking at it from the perspective of saturating neurons or the entropy itself.

- _Improvement over the baselines are minor, which is further aggravated by the fact that analysis was quite limited (see point above)_

	This is a fair criticism in my opinion.

**Reviewer Scores:**

- Reviewer vFaN: They gave the paper a 4, and it is unclear to me whether the additional results would have made the reviewer raise their scores. That being said, I don't think that should be a reason for the paper to be rejected. I suspect the reviewer would have kept their scores because the new "theoretical justification" seems quite light, if that is a theoretical justification at all.
- Reviewer nCj4: They gave the paper a 2. Maybe they would have raised their score to a 4, given the additional experimental results, but I struggle to imagine them flipping their recommendation to acceptance.
- Reviewer UUtk: They gave the paper a 4. It is not clear to me how much the additional results would have changed their assessment, but in principle, I suspect they would have kept a 4.
- Reviewer EQ9A: They gave the paper an 8. They wouldn't have further raised their scores.

---

### Decision · Program_Chairs · 2026-01-26

Reject